# Robustness and innovation in synthetic genotype networks

Javier Santos-Moreno [1,3,4], Eve Tasiudi [2,4], Hadiastri Kusumawardhani [1], Joerg Stelling [2] ✉ & Yolanda Schaerli [1] ✉

Genotype networks are sets of genotypes connected by small mutational changes that share the same phenotype. They facilitate evolutionary innovation by enabling the exploration of different neighborhoods in genotype space. Genotype networks, first suggested by theoretical models, have been empirically confirmed for proteins and RNAs. Comparative studies also support their existence for gene regulatory networks (GRNs), but direct experimental evidence is lacking. Here, we report the construction of three interconnected genotype networks of synthetic GRNs producing three distinct phenotypes in *Escherichia coli*. Our synthetic GRNs contain three nodes regulating each other by CRISPR interference and governing the expression of fluorescent reporters. The genotype networks, composed of over twenty different synthetic GRNs, provide robustness in face of mutations while enabling transitions to innovative phenotypes. Through realistic mathematical modeling, we quantify robustness and evolvability for the complete genotype-phenotype map and link these features mechanistically to GRN motifs. Our work thereby exemplifies how GRN evolution along genotype networks might be driving evolutionary innovation.

A genotype network (also called neutral network)[1–6] is a connected set of genotypes that produce the same phenotype. Within a genotype network, genotypes are directly connected to each other if they differ by a small mutational change. Genotype networks are thought to be a common organizational property of genotype spaces of biological systems at different levels. Ample empirical evidence for their existence supports this notion for RNAs[7–9], proteins[10–14] and binding sites of regulatory proteins[15–18].

In contrast, genotype networks of gene regulatory networks (GRNs) have remained more recalcitrant to investigations. This is an important limitation because GRNs underlie fundamental behavioral and developmental processes[19,20], and because understanding the relationship between a GRN and its biological function (phenotype) is a central area of investigation in modern biology[21]. Current empirical evidence for the existence and features of genotype networks of GRNs

is indirect. It mainly comes from comparative analyses of the expression dynamics and regulatory structure of GRNs in related species showing that rewiring of GRNs during the course of evolution does not necessarily alter the resulting gene expression pattern[22–26]. Indeed, extensive rewiring seems to be very common[27–29]. Complementary theoretical studies revealed that a large number of different GRNs can produce the same phenotype and that many of those GRNs are interconnected by small mutational changes[30–33]. However, these theoretical studies used abstracted models of GRNs not supported by experimental data.

Potential genotype networks for GRNs have three important implications. First, a genotype network of many GRNs can be traversed by making one mutational change at a time without losing the phenotype. Thus, a GRN is robust to those mutations that keep it on the same genotype network[34]. Second, a genotype network implies that

[1]Department of Fundamental Microbiology, University of Lausanne, Biophore Building, 1015 Lausanne, Switzerland. [2]Department of Biosystems Science and Engineering, ETH Zurich and SIB Swiss Institute of Bioinformatics, Basel, Switzerland. [3]Present address: Department of Medicine and Life Sciences, Pompeu Fabra University, 00803 Barcelona, Spain. [4]These authors contributed equally: Javier Santos-Moreno, Eve Tasiudi. ✉e-mail: joerg.stelling@bsse.ethz.ch; yolanda.schaerli@unil.ch

the genotype can evolve while the phenotype is preserved. This is known as phenogenetic drift or system drift[35,36], especially when referring to developmental GRNs. Thereby, genotype networks crucially contribute to evolutionary innovation[6,37]: different genotypes at different positions within the genotype network provide access to genotypes that are part of adjacent genotype networks featuring distinct phenotypes. Consequently, evolving on a genotype network facilitates the exploration of different mutational neighborhoods in genotype space, which may harbor different phenotypes[38,39]. Third, in some cases, the same specific mutation can have different effects depending on the genotype where it occurs: some GRNs, when mutated, retain their phenotype and thus are part of the same genotype networks; the same mutation introduced into other GRNs can lead to mutant GRNs with a distinct phenotype. This common phenomenon, in which the effect of a mutation depends on the genetic background, is known as epistasis[40]. Furthermore, epistasis can itself be dependent on environmental conditions, such as the temperature, the medium, the concentration of an expression inducer, or an enzymatic co-factor[41–43].

Unfortunately, we still have very few experimentally accessible systems that allow us to understand comprehensively how GRNs map to their phenotypes, how they organize into genotype networks, and how these provide robustness to mutations and facilitate access to novel phenotypes[44–47]. To address this issue, here we turned to synthetic biology, which allows us to build GRNs by assembling well-characterized parts that differ by small mutational changes. It also enables us to study GRNs without the common challenges and confounding factors associated with studying GRNs in situ, like the unknown influence of the genetic background, high complexity and interconnectivity of the networks, and pleiotropy of their genes[39,48–50]. Specifically, we decided to build a large set of synthetic GRNs based on CRISPR interference (CRISPRi)[51] using *Escherichia coli* (*E. coli*) cells as host. Our GRNs differ from each other by small mutational changes, thus potentially creating genotype networks. As a starting GRN, we chose a type 2 incoherent feed-forward loop (IFFL-2)[52], which is commonly found in natural systems, including in developmental processes of multicellular organisms such as *Drosophila* blastoderm patterning[53].

Our IFFL-2 has been shown to produce a low-high-low gene expression pattern ("stripe" pattern) across a bacterial population in response to a chemical concentration gradient (Fig. 1a)[51], analogous to the formation of spatiotemporal gene expression patterns guided by morphogen gradients during development[54,55]. In the present study, we characterized a large number of IFFL-2-derived GRNs by incubating bacteria at discrete concentrations of a chemical inducer and evaluating the expression pattern across the concentration range. Overall, we report the construction and experimental exploration of three synthetic genotype networks, each composed of a group of interconnected GRNs and displaying a distinct phenotype, as well as the generalization to the complete genotype-phenotype map using mathematical modeling.

## Results
### A genotype network of GREEN-stripe GRNs
We previously described a synthetic GRN with an IFFL-2 topology that governs the spatial patterning of a population of *E. coli* cells[51]. Our synthetic IFFL-2 responds to a gradient of arabinose (Ara) and produces a stripe of green fluorescence (Fig. 1a). It consists of three nodes connected by repression interactions so that the input node (orange) represses both the intermediate (blue) and the third (green) nodes, and the intermediate represses the third node as well (Fig. 1a). The input node (orange) expression rises with increasing Ara levels. The repression logic produces a concomitant decrease of expression in the intermediate node (blue); consequently, expression of the third node (green) is highest where combined levels of the other nodes are lowest, resulting in a stripe (peak) of gene expression at intermediate Ara concentration (Fig. 1a). In our GRN, this behavior can be easily monitored by fluorescence reporters present in each node: mKO2 (orange), mKate2 (red, here represented in blue for clarity) and sfGFP (green). More specifically, repression is based on CRISPR interference (CRISPRi)[56]. A repressing node produces single guide RNAs (sgRNAs) that recognize specific target binding site (bs) sequences downstream of the promoters in the node to be repressed. CRISPRi-based repression constitutes a versatile framework for synthetic GRN construction due to high programmability and orthogonality and low incremental

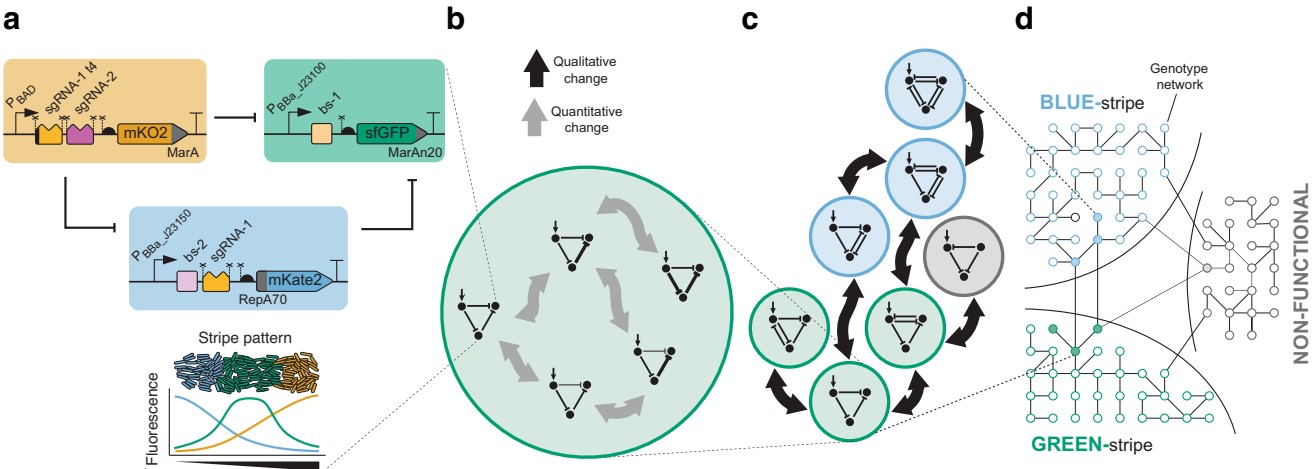

**Fig. 1 | Overview of three interlinked genotype networks of synthetic GRNs built in this study that produce distinct phenotypes for stripe formation: BLUE-stripe, GREEN-stripe, and non-functional.** The different levels of organization are depicted. **a** Details of the molecular implementation of one GRN (GRN 1.1 from Fig. 2), with the resulting GREEN-stripe phenotype schematically depicted below. Key to symbols: Bent arrows: promoters; squares: sgRNA binding sites; jagged rectangles: sgRNAs; crosses: Csy4 recognition sites; semicircles: ribosome binding sites (RBSs); pointed rectangles: reporter genes; and T-s: transcriptional terminators. MarA, MarAn20 and RepA70 denote orthogonal degradation tags. **b** For a GRN topology, there might exist multiple interlinked GRN variants with different parameters that preserve the same phenotype (here shown as connected if they differ by a single change, either qualitative or quantitative). **c** We built several topologies (colored according to the phenotype) that connect within and between genotype networks. **d** The topologies in **c** are part of larger genotype networks that share the same phenotype. Each node in the genotype networks represents one GRN topology with an edge between two nodes if topologies can be interconverted with a single mutational change.

burden[51,57], and we can easily construct new GRN variants with a previously described modular cloning strategy[58].

To construct a synthetic genotype network, we applied two types of changes to GRNs (Fig. 1b, c): qualitative ones, where interactions are gained or lost and thus the topology (i.e. the wiring) of the network changes, and quantitative ones, where the strengths of the regulatory interactions (i.e. the parameters) change[44]. Here, we modified topologies by adding or removing repression interactions, corresponding to a gain or loss of a sgRNA and/or its corresponding binding site. As for parameters, we modulated them in two ways: first, through the choice of three promoters (low, medium, high) that govern transcription of the nodes; and second, by employing four sgRNAs with different strengths. We also used two truncated versions ('t4', truncation of the four 5′ nucleotides) of the sgRNAs, which provides another way to tune repression strength[51]. Overall, changes involve differences ranging from 2-4nt (in the case of promoters and truncated sgRNAs) to 20nt (in the case of the sgRNAs and their binding sites). We consider each of these modifications as a single mutational event (also see **Discussion**) and quantify relations between quantitative and qualitative changes using mathematical models.

Starting from the original GRN (Fig. 2, design 1.1), which produces a GREEN-stripe pattern in a gradient of Ara, we first introduced quantitative changes without modifying the GRN topology. Replacing sgRNA-1t4 with its full-length version to yield GRN 1.2 only slightly decreased the height of the stripe (Fig. 2, design 1.2). A significant increase in the strength of the blue node's promoter in GRNs 1.3 and 1.4 resulted in stripes being asymmetric and shifted towards higher Ara concentrations (Fig. 2, designs 1.3 and 1.4). Thus, the quantitative modifications preserved the GREEN-stripe, but they affected the shape of the stripe.

Previous theoretical work suggested that other GRN topologies populate the same genotype network as our IFFL-2[33,59,60]. To explore those, we next increased GRN complexity by adding one extra repression. The addition of sgRNA-4t4 or of the full-length version sgRNA-4 from the green node to the orange node indeed preserved the GREEN-stripe (Fig. 2, GRNs 2b.1 and 2b.2). Alternatively, we also added a new repression from the blue to the orange node (instead of green to orange) to produce yet a different topology, GRN 2a.1, also displaying a GREEN-stripe phenotype (Fig. 2). Overall, these GRNs demonstrate an uninterrupted genotype network: single (qualitative or quantitative) mutations connect the GRNs, such that distant GRNs are connected by intermediates that preserve the same common phenotype.

## A genotype network of BLUE-stripe GRNs

To explore a different genotype network, we noted that adding a repression from the green node to the blue node in the original GRN (1.1) makes the topology completely symmetrical. In this topology, either the green or the blue node can form a stripe, depending on the parameters. Adding the repression to two different GREEN-stripe GRNs (Fig. 2, GRNs 1.1, 1.4) inverted the roles of the two nodes, producing a BLUE-stripe, with the green node now decreasing with increasing Ara concentration (Fig. 2, GRNs 2c.1, 2c.2, 2c.4). A series of control GRNs confirmed that the GREEN- to BLUE-stripe transition is a consequence of the action of the sgRNA added, and not some spurious context-dependent effect (Supplementary Fig. 1). Thus, a single mutation in some of the GRNs of the GREEN-stripe genotype network suffices to achieve a new phenotype, highlighting the potential of genotype networks for evolutionary innovation.

As already observed for the GREEN-stripe GRNs, quantitative changes in the promoter and/or the sgRNA strengths yielded new BLUE-stripe GRNs in an interlinked network (Fig. 2, GRNs 2c.1 to 2c.8). Many of these GRNs produce skewed BLUE-stripes, but one GRN shows a remarkable symmetry, both in the design and in the phenotype (Fig. 2, GRN 2c.5).

To assess the robustness of the BLUE-stripe genotype network to topological changes, we further increased the complexity to five repression interactions. Adding a new sgRNA from the green to the orange node resulted in GRNs 3.1 to 3.3, all of which produce a BLUE-stripe (shifted to higher Ara concentrations; Fig. 2). Interestingly, this 5-repression BLUE-stripe producing topology can be accessed from a 4-repression topology with either a BLUE-stripe (2 c.6 to 3.1 transition) or a GREEN-stripe (2b.1 to 3.3 transition). The 5-repression GRN 3.2 in turn served as the basis for building a GRN with six repressions between the three nodes – the theoretical maximum number of repressions for a 3-node GRN without self-repression (Fig. 2, GRN 4.1). This design produces a stripe similar to the other ones in its genotype network. In summary, we explored a genotype network of synthetic BLUE-stripe GRNs; these GRNs are connected not only among themselves, but also to the genotype network of the GREEN-stripe GRNs (Fig. 2).

## Ensemble mathematical modeling predicts genotype-phenotype relations

In our synthetic implementations, we explored a large set of functional GRNs (19 3-node GRNs) thanks to the versatility of CRISPRi. Yet, compared to the total number of 1'873'152 possible implementations (42 topologies with 3 to 6 edges, each with 6 possible sgRNAs and 3 promoter efficiencies), this set was too small to conclude quantitatively on robustness and evolvability of genotype networks for GRNs. In addition, some experimental findings required mechanistic interpretations, for example, why we always observed a BLUE-stripe whenever a repression from the green node to the blue node was present. We, therefore, developed mechanistic mathematical models to explore the full genotype space in silico on the one hand, and to analyze mechanisms and guide experimental design on the other hand.

The models represent detailed interactions between Ara-controlled gene expression, sgRNA and fluorescent protein expression, dCas9-sgRNA interactions, CRISPRi-mediated control of gene expression, and component degradation or dilution, as illustrated in Fig. 3a for an IFFL-type 2 GRN. To capture the experiments conducted in microplate readers realistically, we included simplified model components for microbial growth and fluorescent protein maturation into the dynamic (ordinary differential equation-based) models; the overall model structure allowed us to specify all possible topologies and parametrizations (see **Methods** for details). For model calibration, we used experimental data for two-node GRNs with different sgRNAs[51], and for a selection of three-node GRNs (Supplementary Fig. 2). Importantly, we constrained parameters to map 1:1 to biological components, such that a specific sgRNA or promoter had the same value for, e.g., affinity constants, throughout all models (see **Methods**). In principle, this enables predictions for the entire genotype space of GRNs implementable with the synthetic parts used.

The models faithfully predicted the experimentally observed behaviors of GRNs that differed in topologies or parts from those used in model calibration (i.e., the independent validation set), as shown in Fig. 3b, c and Supplementary Fig. 2. In addition, the inferred sgRNA affinities were qualitatively consistent with measured repression strengths (Supplementary Data 1). Overall, we conclude that the mathematical models are sufficiently realistic for comprehensive in silico analyses.

## Models enable a comprehensive analysis of robustness and evolvability

To assess the global genotype-phenotype (GP) map, we predicted the behavior of all 1'873'152 possible GRNs and established three phenotypes for stripe formation: BLUE-stripe, GREEN-stripe, and nonfunctional (NF; see **Methods** for definitions). For the genotypes, each parametrized model (i.e., representing a GRN) is a genotype and a single mutation amounts to a change in one parameter value. Note that such a parametric change may also alter the topology, for

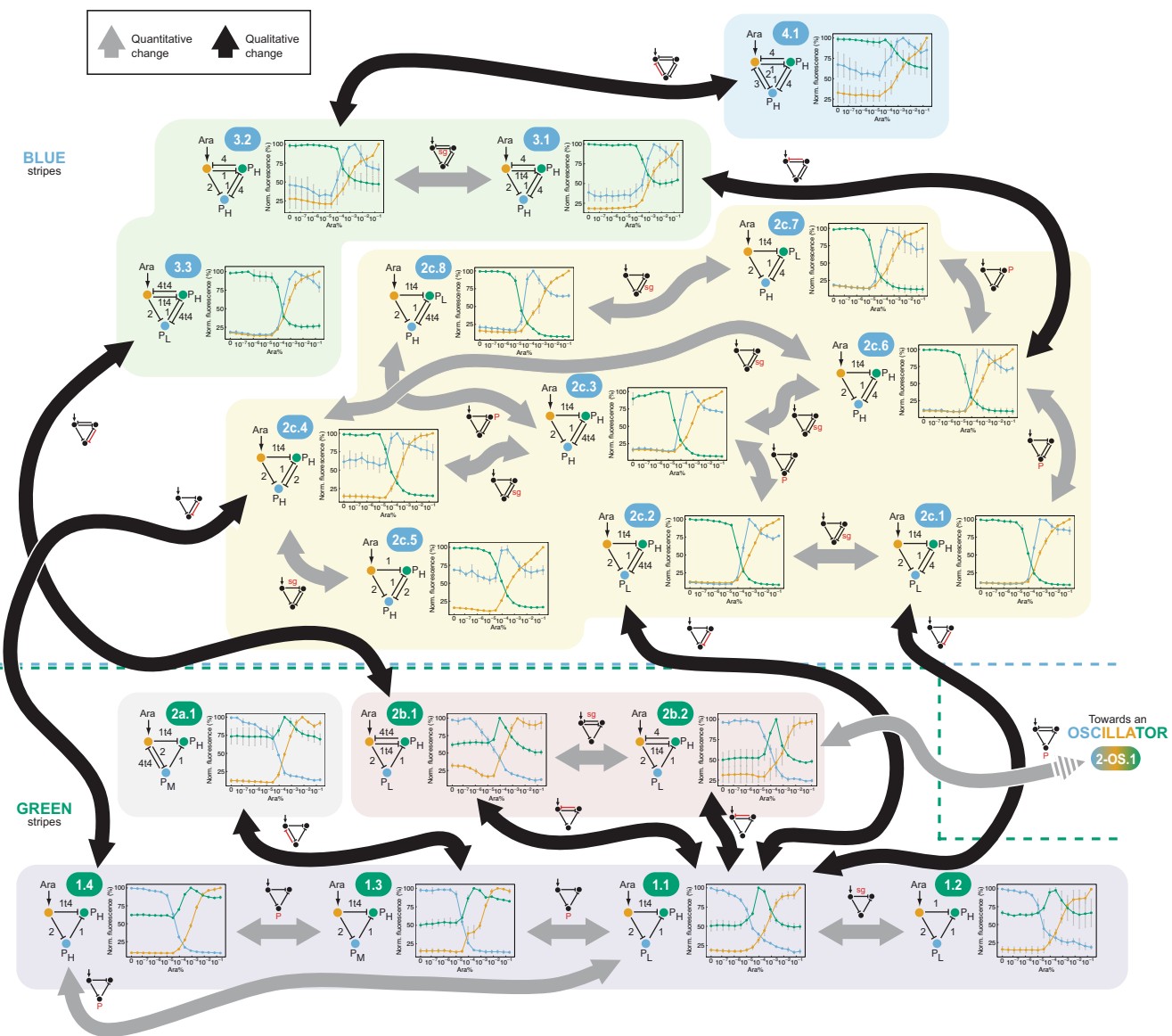

**Fig. 2 | A synthetic genotype network of GREEN-stripe (bottom) and BLUE-stripe (top) GRNs.** Starting from a CRISPRi-based GREEN-stripe incoherent feed-forward loop (IFFL-2, design 1.1), we introduced quantitative (gray) or qualitative (black) changes to produce a genotype network of synthetic GREEN-stripe GRNs. Single changes enabled the transition from GREEN-stripe GRNs to BLUE-stripe GRNs. We then explored this BLUE-stripe genotype network through parameter (gray) or topology (black) changes, up to the GRN with the maximum number of repressions for a 3-node GRN without self-repression, GRN 4.1. Dashed lines represent the boundaries of the GREEN-stripe and BLUE-stripe genotype networks. Each design is denoted with a unique code displayed in an ellipse (color-coded according to the phenotype), with the details of the design provided in the diagrams below. The GRN code denotes both the complexity (i.e. number of interactions) and the topology, as well as the specific implementation. A higher starting number of the GRN (e.g. 2 vs. 1) reflects an increased complexity, while different

topologies sharing the same complexity are denoted with different letters (e.g. 2a vs. 2b). Lastly, the final number (e.g. 2b.1 vs. 2b.2) distinguishes specific implementations within the same topology. Numbers by the repressions indicate the identity of the sgRNAs, while $P_H$, $P_M$, and $P_L$ represent constitutive promoters (BBa_J23100, BBa_J23102, and BBa_J23150, respectively). Small GRNs by the arrows display the changes (in red) between the connected GRNs: topology changes (T) or changes in the promoter strength (P) or in the identity of the sgRNAs (sg). The phenotype of each GRN was characterized in a microplate reader by determining the fluorescence of the three protein reporters (mKO2 (orange), sfGFP (green), and mKate2 (red, here depicted in blue for clarity)) as a function of arabinose (Ara) concentration. Data show the mean of three biological replicates, with error bars depicting the s.d. of normalized replicates ($n = 3$). GRNs sharing a common topology are grouped with a colored background. The dashed arrow indicates the transition towards another synthetic genotype network (that of oscillators, Fig. 7).

example, by effectively eliminating a repressive interaction. The summary in Fig. 4a shows that functional GRNs (GREEN- and BLUE-stripe GRNs) are relatively rare (80% NF, 10% BLUE-stripe, and 10% GREEN-stripe), both in terms of topologies and of GRNs within a topology, consistent with previous results for RNAs and proteins[1]. In more detail, as suggested by our experimental data, mutual inhibition of green and blue nodes is required to enable a topology that can produce two different phenotypes – either BLUE- or GREEN-stripes, depending on the parameters.

Surprisingly, we found that the BLUE-stripe, GREEN-stripe, and NF genotype networks were each composed of a single connected network (a strongly connected component in graph terms). Hence, it is possible for very different GRNs to establish the same phenotype and to reach any of these GRNs in single mutations without changing the phenotype. To analyze the GP map in more detail, we first determined genotype evolvability, defined as the number of phenotypes accessible from a GRN in a single mutation change (i.e., in the 1-neighborhood of a GRN in the genotype network)[61]. Most functional GRNs were adjacent

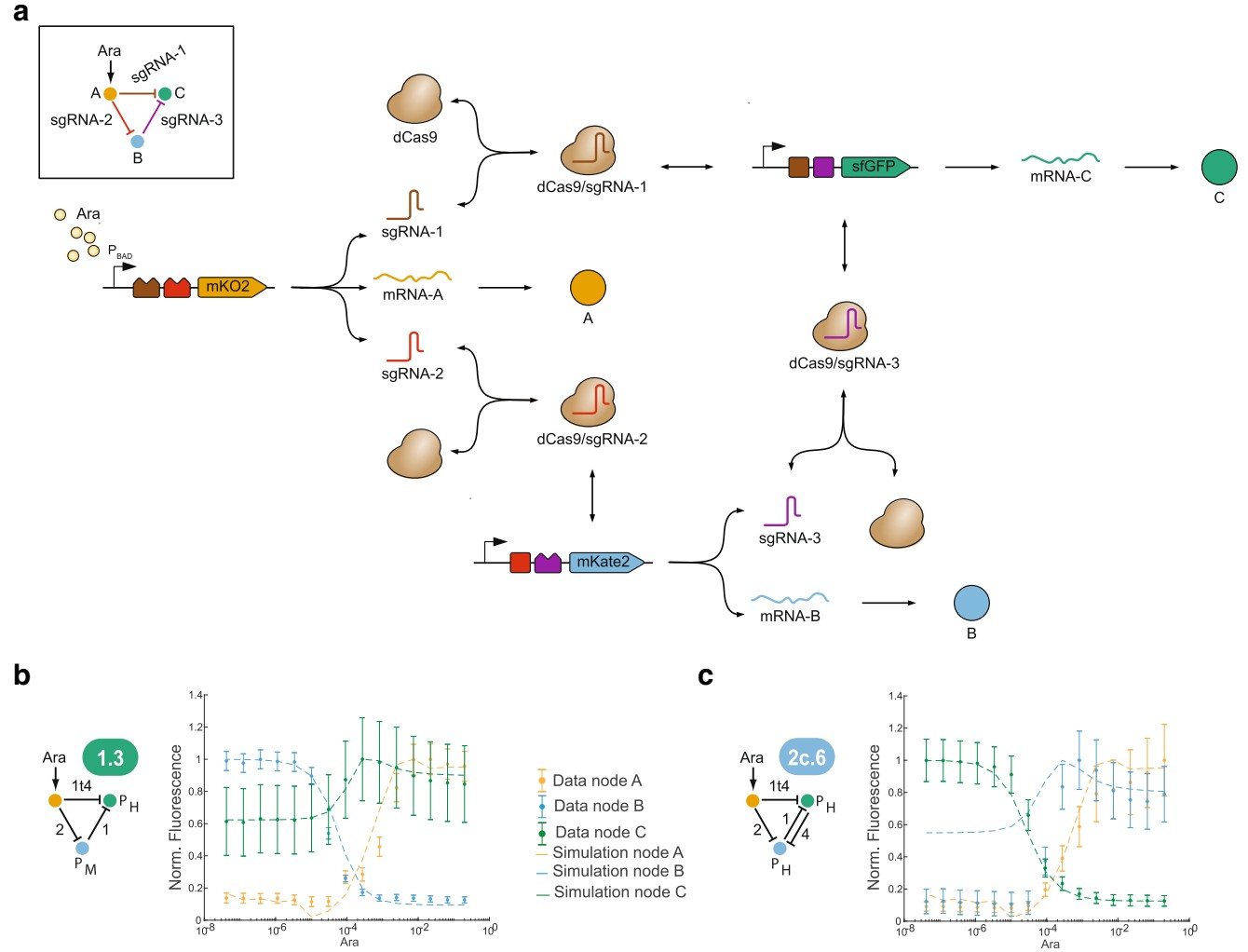

**Fig. 3 | Model overview and predictions. a** Model components and processes for the example of an IFFL-type 2 using generic sgRNAs 1-3 (inset). Arabinose induces expression of sgRNA-1 (brown jagged rectangle), sgRNA-2 (red) and A (orange). sgRNA-1 and sgRNA-2 form complexes with dCas (brown), and A undergoes translation (mRNA-A) and maturation (A). sgRNA-3 (purple), B (blue) and C (green) are constitutively expressed (bent arrows). sgRNA-3 forms a complex with dCas, and B and C are translated (mRNA-B, mRNA-C) and mature (B,C). sgRNA/dCas complexes can bind to their target sites (brown, red, and purple squares) and inhibit gene expression. RNAs are subjected to active degradation, proteins (A, B, C) to active degradation and dilution, and complexes to dilution. **b**, **c** Independent model predictions (dashed lines) compared to experimental data (symbols, as in Fig. 2) for two example GRNs (Supplementary Fig. 2). Models were adapted to account for the sgRNAs in the implemented GRNs. Symbols show the mean of three biological replicates, while error bars show s.d. based on error propagation ($n = 3$).

to at least one GRN of all the other phenotypes, in contrast to the NF GRNs (Fig. 4b), indicating the high evolvability of functional GRNs specifically. However, this adjacency does not imply that it is likely that a mutation will yield a new (specific) phenotype, for example, when this phenotype is rare in the neighborhood, as illustrated in Fig. 4c. We, therefore, analyzed possible transitions between phenotypes (Fig. 4d). Entries on the diagonal show that NF GRNs are highly robust to mutations, and functional ones moderately robust. Transitions to a NF phenotype dominate for all GRNs, but switching between a BLUE- and GREEN-stripe has a substantial probability. To make these results more interpretable, we used random walks on the genotype network to determine the mutational path lengths for altering phenotypes (see **Methods**). Median path lengths were 2 for functional GRNs and 12 for NF GRNs (Supplementary Fig. 3), representing a high evolutionary barrier to establish function, but high evolvability of function. Importantly, when we repeated the analysis with additional artificial inhibition and promoter strengths (see **Methods**) as a control, the results were very similar (Supplementary Fig. 4).

To relate this analysis to our synthetic biology approach in more detail, we determined robustness to changes in GRN parts and interactions. As in prior work[62], we define robustness of a phenotype (GREEN-stripe, BLUE-stripe, or non-functional) of a reference GRN by the fraction of neighboring GRNs that have the same phenotype when we apply a single mutation (i.e., changing an inhibitory interaction or a promoter strength). Over all genotypes, phenotypes were most sensitive to alterations of promoter strengths for the blue and green nodes, and (in a bimodal manner) of sgRNA-mediated interactions between these two nodes (Fig. 4e for the BLUE-stripe phenotype and Supplementary Fig. 3 for the GREEN-stripe and NF phenotypes). To test these predictions, we asked how we could convert the phenotype of topologies that theoretically afford both functional phenotypes, but for which we experimentally observed only BLUE-stripes. Specifically, we used the models to predict modifications of GRNs in our experimentally implemented BLUE-stripe genotype network that yield a GREEN-stripe. The requirements to swap between the functional phenotypes mirrored the parameter sensitivities (Fig. 4e), namely promoter and inhibition strengths between the blue and green nodes. Similar to Munteanu et al.[63], we predicted that a stronger inhibition (higher sgRNA affinity) on the green node with the correct promoter efficiencies for both green and blue nodes was enough to alter the

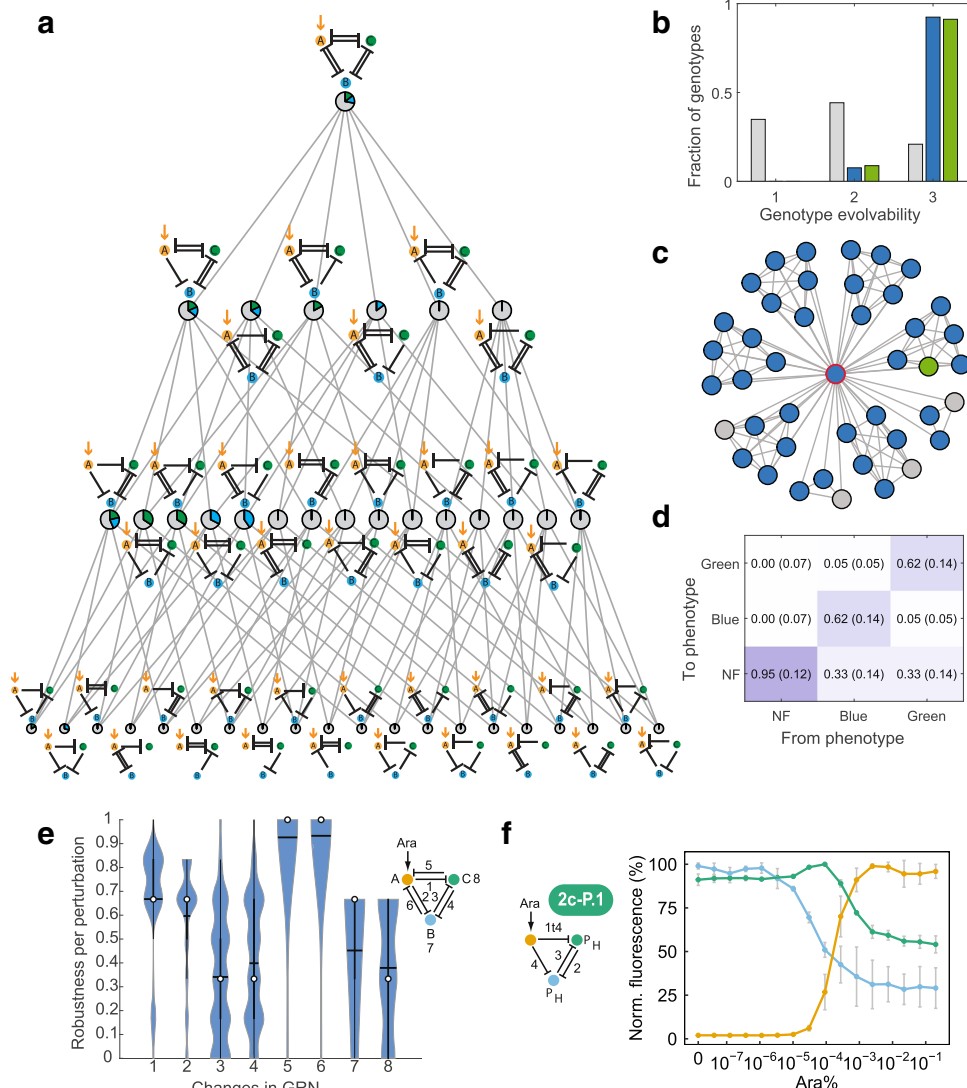

**Fig. 4 | Model-predicted robustness and evolvability. a** Network topologies (schemes as in Fig. 1) and their neighbor relations (grey lines indicating that GRNs in two topologies are reachable by a single mutation). Pie charts indicate fractions of GRNs (genotypes) with BLUE-stripe (blue), GREEN-stripe (green) and non-functional (NF; grey) phenotype per topology. **b** Distributions of evolvability (number of phenotypes in the 1-neighborhood of a genotype) for genotypes with BLUE, GREEN, and NF phenotype (colors as in **a**). **c** Example subgraph for the red-circled node with evolvability 3 and its direct neighbors (37 BLUE-, 1 GREEN-stripe and 4 NF, colors as in **a**). **d** Transition frequencies (median, interquartile range in brackets) between indicated phenotypes resulting from transitions between neighboring genotypes. **e** Robustness (fraction of neighboring genotypes with the same phenotype as a given genotype) classified by type of perturbations (indicated by numbers in GRN diagram), namely changes of sgRNA type (1–6) and promoter strengths (7–8) for the respective interactions and nodes for GRNs with BLUE-stripe phenotype. **f** Experimental test of a model prediction to convert a BLUE-stripe GRN to a GREEN-stripe GRN with the same topology; symbols as in Fig. 2. Data show the mean of three biological replicates, with error bars depicting s.d. of normalized replicates (*n* = 3).

phenotypes. In experiments, GRN 2c-P.1 indeed showed a phenotype switched to GREEN-stripe, as illustrated in Fig. 4f. Thus, our models provide global as well as detailed, experimentally actionable, analyses of GRN robustness and evolvability.

**Combining IFFL and mutual inhibition enhances evolvability**
To understand mechanisms in biological networks, a powerful approach is to focus on the topology, and specifically on network motifs, small sub-networks that are critical for function, and reveal design principles[64]. For example, they helped investigate robustness and (to a certain extent) evolvability of the gap gene patterning system of insects[65]. However, a systematic analysis of GP maps in terms of topologies and motifs is non-trivial because, for example, here the number of GRNs per topology varies between 3'456 and 746'496.

We propose to address this challenge by computing genotypic robustness and evolvability in two ways: (i) for the entire 1-neighborhood of each GRN and (ii) for only those neighbors of a GRN that have the same topology. The results for evolvability in Fig. 5a show a clear differentiation of topologies into five clusters (see also Supplementary Table 2). Topologies that are always non-functional (evolvability of one with fixed topology) can only become functional when evolving (directly or indirectly) to topologies containing an IFFL. Evolvability increases further by incorporating a mutual inhibition (MI) of blue and green nodes, complementing our earlier observations. Importantly, these network motifs, and not GRN complexity (i.e. number of interactions) determine evolvability.

Regarding robustness (Fig. 5b for the GREEN-stripe phenotype, and Supplementary Fig. 2), functional GRNs – those with an IFFL – perform substantially better (have higher robustness in both

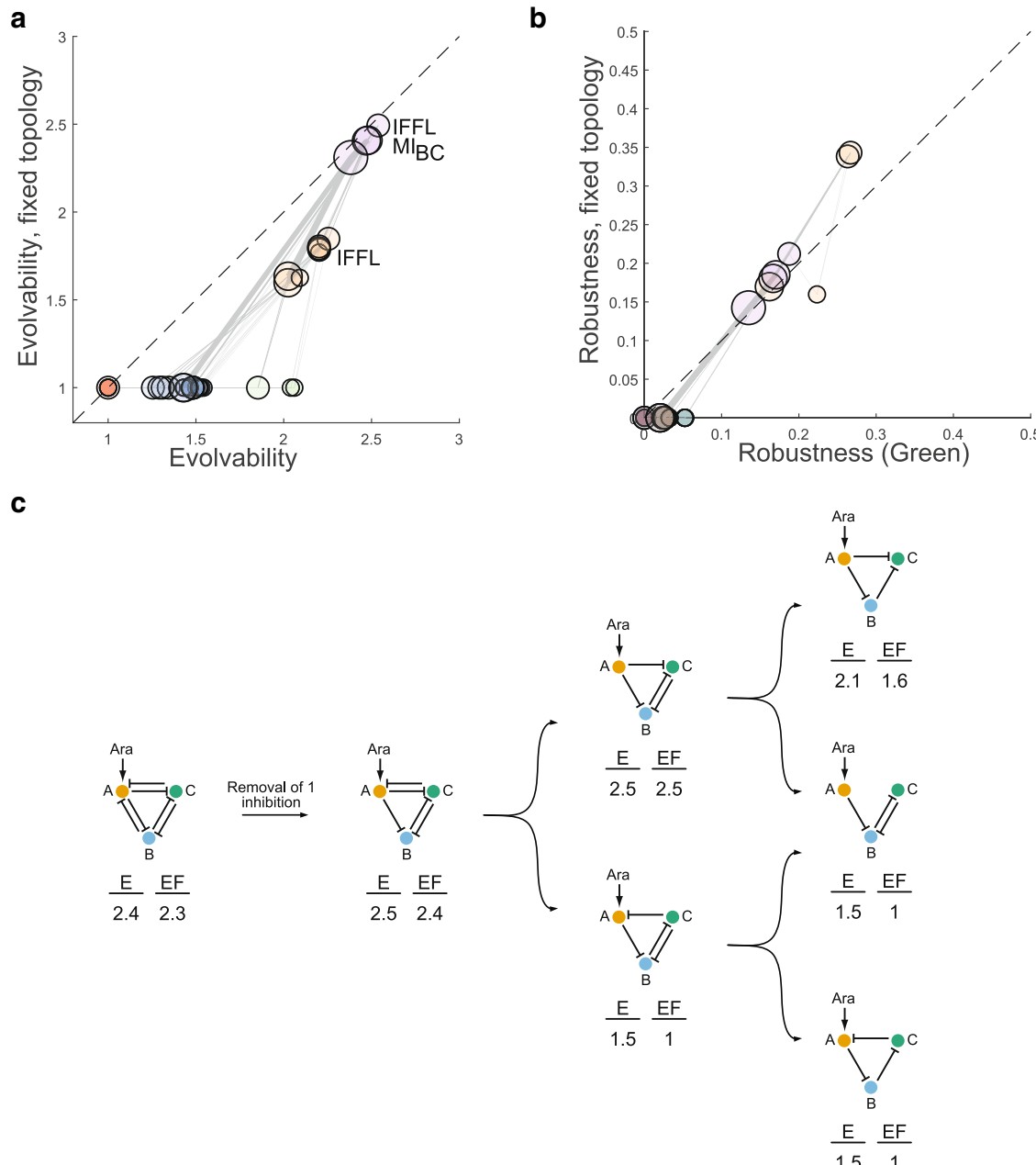

**Fig. 5 | Mechanistic basis of robustness and evolvability. a** Average predicted phenotypic evolvability per topology (circles) over all neighboring genotypes (x-axis) vs only genotypes with the same, fixed topology (y-axis). Topologies were clustered (indicated by colors; k-means clustering with k = 5). Circle sizes reflect numbers of repressive interactions in a topology and grey lines link adjacent topologies as in Fig. 4a. The dashed diagonal indicates equal evolvability with and without fixed topology. $MI_{BC}$: mutual inhibition of nodes B (blue) and C (green). **b** Quantification of robustness for the GREEN-stripe phenotype; results shown as in **a**, with the same coloring of clusters. **c** Example for the effect of transitions between topologies (arrows: removal of one inhibitory interaction) on average evolvability (E) and evolvability with fixed topology ($E_F$) measured as in **a**.

dimensions) than the other GRNs. In contrast to evolvability, including the MI motif decreases the robustness of a given functional phenotype, again not correlated with GRN complexity. This is consistent with a previously observed trade-off between genotypic evolvability and robustness[66], derived from abstractly modeled GRNs. We argue that experimentally validated GRN models and their analysis via network motifs provide a more realistic and mechanistically interpretable view, as illustrated in Fig. 5c.

**Epistatic interactions within the BLUE-stripe genotype network**
A third aspect important for evolution is the existence and prevalence of epistatic interactions; small sample sizes make the latter difficult to

estimate in vivo, but a recent analysis in yeast showed a prevalence of ~3%[67]. To relate to genotype networks, we focused on one type of epistasis, namely where two sequential changes ('A' and 'B') allow genotype network exploration without a phenotype loss, while the same changes in the reverse order (first 'B' then 'A') goes along with a loss of the phenotype after the first change (Fig. 6a). Our models predict median prevalences of ~4.5% for functional GRNs, and a lower one for non-functional GRNs (~2.6%) (Fig. 6b). This is qualitatively consistent with prior experimental studies; it highlights again the need to differentiate between functional and non-functional phenotypes.

In our synthetic GRNs, we also discovered that the order of the changes actually matters. We found an instance of epistasis in the

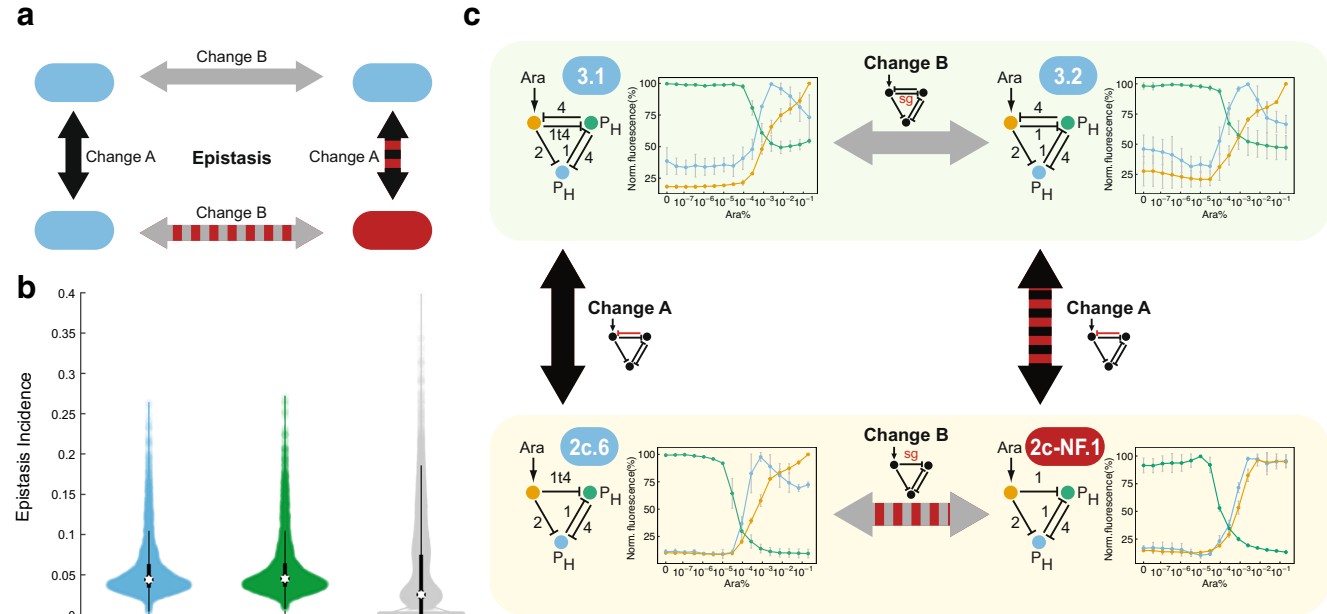

**Fig. 6 | Epistatic interactions. a** Schematic illustration of epistasis. Two consecutive changes 'A' and 'B' allow for a smooth 'walk' within the BLUE-stripe genotype network. However, the same changes in the reverse order transition through a GRN that is in the non-functional genotype network. **b** Model-based estimation of epistasis incidence by sampling 5% of pairs of GREEN and BLUE, and 2% of NF GRNs within their corresponding genotype network. Circles: median; vertical lines: interquartile range. **c** Experimental demonstration of epistasis. The transition from

GRN 2 c.6 to 3.2 within the BLUE-stripe genotype network requires two changes in a specific order, via intermediate 3.1. The reverse order of changes involves a non-functional intermediate, 2c-NF.1. Grey arrows: parameter changes; black arrows: topology changes; red shading: connections from/to non-functional GRNs. Data show the mean of three biological replicates, with error bars depicting s.d. of normalized replicates ($n = 3$).

transition from topology 2c to topology 3. Starting from GRN 2c.6, the addition of sgRNA-4 from the green to orange node yielded the functional GRN 3.1, which could then be modified by replacing sgRNA-1t4 with its stronger repressing full-length version to produce GRN 3.2, also functional (Fig. 6c). However, the same changes in the reverse order required traversing through a non-functional intermediate, GRN 2c-NF.1, implying that strengthening the repression from the orange to the green node was only tolerated if an opposing interaction was present. These findings underline the usefulness of combining synthetic GRNs and modeling to analyze genotype networks comprehensively.

### A genotype network of oscillating GRNs

Finally, to evaluate if our findings on the existence of genotype networks translate to other definitions of phenotypes, we exploited prior theoretical work. It demonstrated that GRNs with topologies like GRN 2b.2 can produce temporal oscillations[65,68–70] – its topology actually contains that of the repressilator, a well-known molecular oscillator that relies on a time-delayed negative feedback structure[71]. In addition, we previously demonstrated that CRISPRi can be employed to build a repressilator – a circuit that we named the CRISPRlator[51]. The CRISPRlator is composed of three nodes forming a negative feedback loop, each expressing a sgRNA that represses the next node in the loop. An increased level of any given node triggers a cascade of repression interactions that eventually bring its own levels down again, leading to oscillations.

Starting from GRN 2b.2, a single parameter change (a stronger promoter in the blue node) sufficed to produce an oscillatory phenotype, as assessed through a continuous characterization of the gene expression dynamics in a microfluidic device (Fig. 7a, GRN 2-OS.1). The removal of sgRNA-1t4 did not abolish oscillations and indeed rendered the topology closer to that of the classical repressilator; substituting the reporters with ones better suited for our microscopy settings also

preserved the oscillatory dynamics (Figs. 7a, 1-OS.1 and 1-OS.2). Finally, the replacement of P_BAD with a constitutive promoter (P_H) recreated the previously reported CRISPRlator displaying highly regular oscillations (Fig. 7a, 1-OS.3). Hence, a genotype network composed of synthetic GRNs that enables exploration of the genotype space without a phenotype loss exists also for dynamic phenotypes.

By adjusting our mathematical model ensemble to account for dynamic phenotypes (by parametrizing dilution rate and arabinose concentration in the microfluidic device; see **Methods** for details), we could predict which topologies oscillate. As expected, most topologies that contained a repressilator were able to show oscillations (Fig. 7b, Supplementary Table 3). Intriguingly, akin to the stripe-forming networks (e.g., Figs. 4a, 5b), not necessarily the most complex topology had the highest fraction of oscillating GNRs (Fig. 7b), which can point to trade-offs between genotypic evolvability and robustness. As for the stripe phenotypes (Fig. 4d), we next analyzed the transition probabilities between the oscillatory and NF phenotypes. The oscillatory phenotype is quite robust to small mutational perturbations (Fig. 7c); yet obtaining an oscillatory phenotype from an NF phenotype is extremely rare. Note, however, that we obtained these results with minimal re-parametrization of the models: this demonstrates the consistency of the models with different phenotypes of interest, but we did not explicitly validate the models on oscillatory experimental data.

### Discussion

Synthetic biology has recently been used to address fundamental questions in the molecular evolution of regulatory networks[39,72–75]. Taking advantage of synthetic biology tools allowed us to assess several key features of GRN genotype networks experimentally and theoretically. We demonstrate that distinct synthetic GRNs can indeed be part of the same genotype network in which small mutational changes enable the smooth transition between any two GRNs without losing the target phenotype. On the other hand, genotypes very close in the

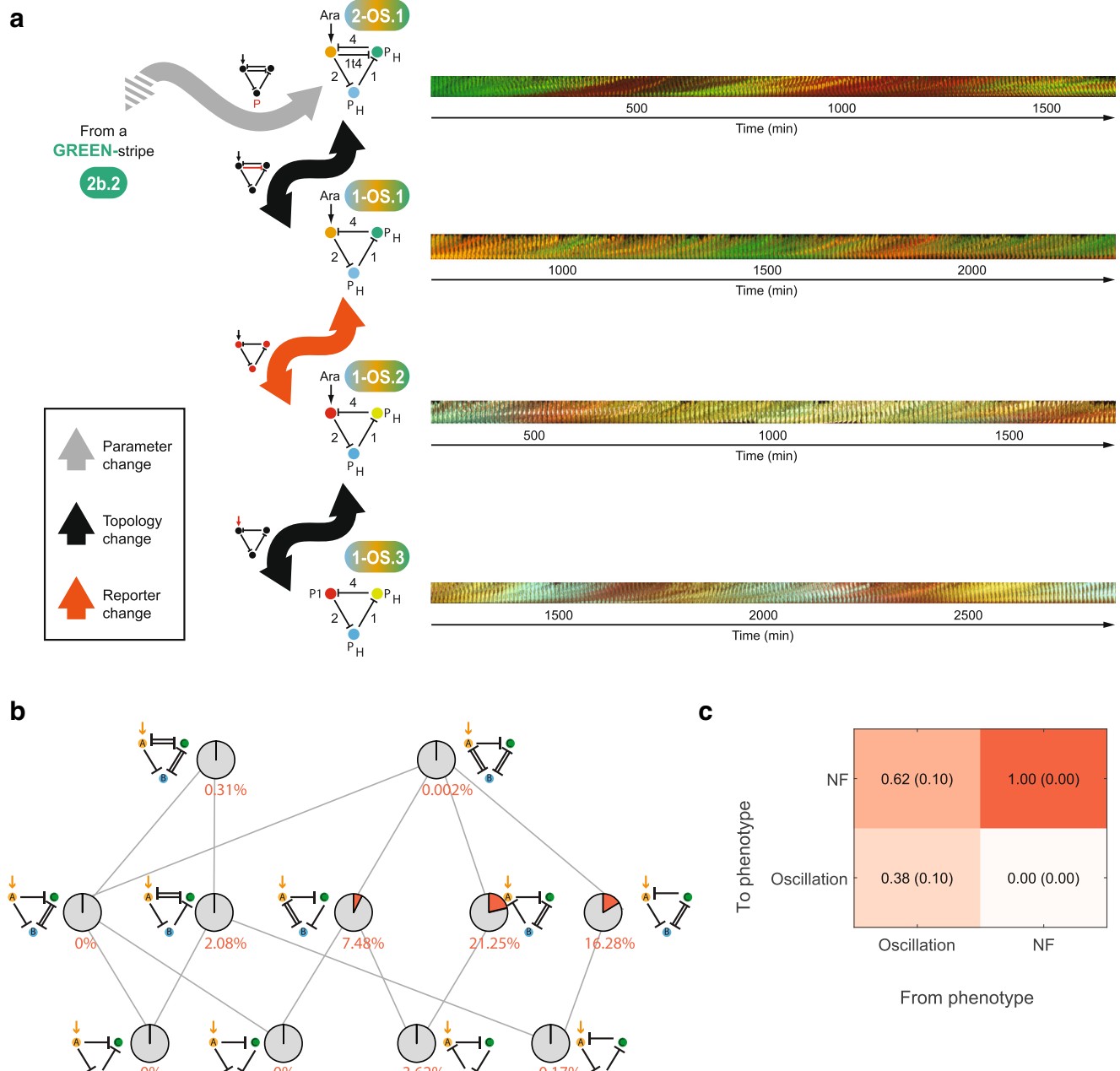

**Fig. 7 | A synthetic genotype network of oscillatory GRNs. a** Starting from the GREEN-stripe GRN 2b.2, a stronger promoter for the blue node leads to a GRN that displays oscillatory expression. Few changes suffice to transition to the CRISPRlator[51], with all intermediates showing the oscillatory phenotype. Bacteria carrying the indicated GRNs were grown in continuous exponential phase in a microfluidic device and imaged every 10 min for up to two days. Oscillations are shown as kymographs: images of the microfluidic chamber (hosting ~100 cells) are displayed in order in a timeline montage. Due to an overlap of mKO2 and mKate2 in the same channel, red color for 2-OS.1 and 1-OS.1 (top) represents both mKO2 and mKate2, while green corresponds to sfGFP. Different fluorescent reporters mCherry (red), mCitrine (yellow) and Cerulean (blue) allowed us to characterize the dynamics of all three nodes (1-OS.2 and 1-OS.3; bottom). **b** Part of network topologies (schemes as in Fig. 1) and their neighbor relations predicted by the mathematical framework. Similar to Fig. 4a, the pie charts indicate fractions of GRNs (genotypes) with oscillatory (orange) and non-functional (NF; grey) phenotypes per topology. Numbers: percentages of occurrences of the oscillatory topologies. **c** Similar to Fig. 4d, transition frequencies between the neighboring indicated phenotypes.

mutational distance can have distinct phenotypes, and genotypes in different positions of the genotype network can access different novel phenotypes. More specifically, our results conform to a 'bowl of spaghetti' metaphor known from the RNA world[1,76], but with subtle differences in the robustness and evolvability of GRN function. Both the connectedness of genotype networks and that they spread throughout the genotype space were proposed to be crucial for robustness and evolutionary innovation[77]. Our study provides experimental as well as theoretical evidence for it. Importantly, this is not restricted to stripe-

forming GRNs. For example, our data suggests an evolutionary trajectory from a GREEN-stripe to an oscillatory GRN.

Prior theoretical work on GRNs highlighted that the definition of mutations is crucial for the analysis of robustness and evolvability, but it used abstract concepts such as signal-integration logic[66]. Here, quantitative changes involve mutations in DNA sequences encoding both *cis*-regulatory elements (promoters) and *trans*-regulatory factors (sgRNAs). The three constitutive promoters used in this study differ in 2–4 nucleotides. Mutations in *trans* involve a 4 nt difference between

full-length or truncated sgRNAs or changing the DNA-binding part of the sgRNA (20 nt). Such small differences in *cis*- or *trans*-regulatory regions have been found to play a role in the evolution of natural networks[78]. Qualitative changes between our GRNs involve the gain or loss of a sgRNA and its corresponding binding site. Comparative studies show that rewiring within transcriptional networks is common in natural systems, often facilitated by duplication events and subsequent diversification[79]. Our transitions in genotype space include duplications of a regulatory element (e.g., from GRN 1.4 to 2 c.4: a new sgRNA-2 in the green node binds to the existing bs-2 in the blue node), binding site duplications (e.g., from GRN 2 c.6 to 3.1: the existing sgRNA-4 in the green node binds to a new bs-4 in the orange node), and combinations of both (e.g., from GRN 1.1 to 2b.2: a new sgRNA-4 in the green node binds to a new bs-4 in the orange). Hence, our synthetic GRNs and their mathematical models – while not representing individual mutations at the DNA level – incorporate realistic single mutational changes in evolutionary trajectories. Future work could extend this approach to investigate the effect of varying gene numbers on robustness and evolvability, as in prior work with abstract GRNs[62].

Another critical aspect is the selection of phenotypes, and correspondingly the phenotypic space. Our GRNs displayed phenotypes – stripe formation in a gradient and oscillations – that are crucial for metazoan development of many organisms and body structures[54,55]. We used coarse-graining (e.g., BLUE-stripe, GREEN-stripe, and NF) to characterize these phenotypes. It is a justified simplification and common practice when studying the evolution of GRNs[33,39,80,81]. However, because the underlying experimental and simulation data is quantitative, our coarse-graining is already a step towards a more realistic scenario than Boolean logic behaviors[64,82]. Correspondingly, GRNs in one of our genotype networks may display quantitatively different behaviors. In a natural system, many of these differences could be buffered downstream[83], while others might lead to different organismal features. Coarse-graining to only three phenotypes for stripe-forming GRNs, however, implied that we could not study phenotypic robustness and evolvability. They are suggested to exhibit subtle, yet important differences from their genotypic counterparts[37] and could be addressed in future studies.

We combined experiments and modeling to characterize the GP map of our synthetic GRNs realistically and comprehensively; neither of the two approaches alone could achieve this. It allowed us to link robustness and evolvability to small network motifs. Even when embedded in larger networks, motifs are necessary and sufficient for many network functions[64]. For example, one of the best-studied GRNs, the gap gene network in insects, comprises several genes that regulate each other by intricate repression motifs (among them IFFL-2) to produce stripes of gene expression that eventually establish anterior-posterior embryo segmentation[26]. Comparative analysis in different dipteran insects provides strong evidence for system drift and evolution on a genotype network[26,44,84]. Interestingly, *Anopheles gambiae* and *Drosophila melanogaster* have inverted stripe expression patterns of the gap genes *giant* (gt) and *hunchback* (hb)[26,85]. Given the potential for simple mutational changes to switch between GREEN- and BLUE-stripe in IFFL-2-based synthetic GRNs, our work provides possible evolutionary paths for this role-switching in dipteran gap GRNs.

Similarly, the network topology of 2-OS.1 is known as AC-DC[69,70]. It is involved in patterning the vertebrate neural tube[86] and in the *Drosophila melanogaster* gap gene network[65]. Importantly, the AC-DC circuit may explain how segmentation of the developing embryo in long germ-band insects such as *Drosophila*, where all segments are determined simultaneously, could have evolved from short germ-band insects, where oscillations establish the segments sequentially[87]. Our synthetic GRNs illustrate that it is indeed straightforward to transition between stripe-forming and oscillatory networks.

More generally, epistasis profoundly complicates our understanding of how GRNs respond to mutations and is therefore studied extensively in different biological systems across scales[40]. So far, most work on the mechanistic causes of epistasis in GRNs has been purely computational[80,88–90]. These studies suggest that epistasis is common in GRNs and that it relies on nonlinear mechanisms of gene regulation, e.g. those generated by mutual repressions and feedbacks. Because experimental validation of these predictions is largely missing, mainly due to the high complexity of natural GRNs, our results underscore that small, mechanistically well-understood (synthetic) GRNs are promising model systems to study mechanisms and prevalence of epistasis[39,41,75,91]. The epistatic relation between GRNs 2 c.6 and 3.2 also exemplifies how hybrid incompatibilities could arise: certain crossings between individuals carrying these GRNs could lead to GRNs without a stripe phenotype. This is consistent with system drift being an important source of hybrid incompatibilities that cause reproductive isolation[92–94].

Overall, we demonstrated that building many synthetic counterparts of natural systems, combined with realistic mathematical models to explore complete GP maps, can provide us both with general principles and specific insights into GRN evolution and function. We, therefore, anticipate that synthetic biology will gain further relevance in deciphering the mechanisms of molecular evolution.

## Methods

### GRN construction

Genes encoding sfGFP, mKO2, mKate2, mCherry, mCitrine, Cerulean, Csy4, and dCas9 were obtained as previously described[51,58]. All reporters were fused to orthogonal degradation tags[95], as follows: mKO2-MarA, sfGFP-MarAn20, RepA70-mKate2, mCherry-MarA, mCitrine-MarAn20, RepA70-Cerulean. Primers were purchased desalted from Microsynth or Sigma-Aldrich. The GRNs were constructed employing a previously described Gibson-based cloning framework that allows for the fast and modular cloning of synthetic gene networks[58]. Briefly, the method consists of two steps: step 1 involves Gibson assembly of transcriptional units into individual intermediate plasmids; in step 2, these plasmids are digested with restriction enzymes so that the resulting flanking regions contain overlaps that drive a second Gibson assembly into a single plasmid to yield the final GRN. For step 1, all DNA parts carried the same Prefix (CAGCCTGCGGTCCGG) and Suffix (TCGCTGGGACGCCCG) sequences for modular Gibson assembly using MODAL[96]. Basically, forward and reverse primers annealing to Prefix and Suffix sequences, respectively, were used for PCRs that added unique linkers to the DNA parts. PCR amplifications were column-purified using the Monarch PCR & DNA Cleanup Kit (NEB), and assembled using NEBuilder HiFi DNA Assembly Master Mix (NEB, 1 h 50 °C) into backbones previously digested with corresponding restriction enzymes (NEB, 1 h 37 °C) to yield intermediate plasmids containing individual transcriptional units. In step 2, these intermediate plasmids were digested with enzyme sets yielding overlapping sequences, purified, and assembled as described above. 1 µl of non-purified Gibson reaction was transformed into 50 µl of electro-competent NEB5α cells, and 2/5 of them were plated onto selective agar plates. Plasmids used in this study are listed in (Supplementary Table 1) and all sequences are provided in a supplementary file (Supplementary Data 2).

### Microplate reader experiments

Gene expression of fluorescent reporters was used to assess synthetic GRN performance; fluorescence was measured in microplate readers (except for the microfluidic experiments in Fig. 7). MK01[97] electro-competent cells were transformed with a "constant" plasmid encoding proteins required for GRN function (namely dCas9 and Csy4)[58] as well as with a "variable" vector bearing AraC (when needed) and the CRISPRi GRN (see Supplementary Table 1). 2 ml of selective LB were inoculated with single colonies and incubated at 37 °C for ~6 h with 200 rpm shaking; cells were pelleted at 4000 rcf and resuspended in

selective EZ medium (Teknova) containing 0.4% glycerol. 120 μl of 0.05 $OD_{600}$ bacterial suspensions were added per well on a 96-well CytoOne plate (Starlab), and 2.4 μl of L-arabinose (Sigma) were added to yield the indicated concentrations. Plates were incubated at 37 °C with double-orbital shaking in a Synergy H1 microplate reader (Biotek) running Gen5 3.04 software. Fluorescence was determined after ~16 h with the following settings: mKO2: Ex. 542 nm, Em. 571 nm; sfGPF: Ex. 479 nm, Em. 520 nm; mKate2: Ex. 588 nm, Em. 633 nm. Fluorescence levels were treated as follows: i) the fluorescence signal in a blank sample was subtracted, ii) the resulting value was divided by the absorbance at 600 nm to correct for differences in bacterial concentration, and finally iii) the bacterial auto-fluorescence of a strain with no reporter genes was subtracted. Subsequently, corrected fluorescence was normalized to a percentage scale by dividing all values of a given color by the highest value of that color. Normalized data were plotted in R[98] using RStudio 1.0.143 (running R 3.4.0). Source data are provided in a supplementary file (Source Data).

### Microfluidic experiments

MK01[97] electrocompetent cells were transformed with the constant plasmid pJ1996_v2[58] and a variable plasmid encoding a CRISPRi GRN. Single colonies were used to inoculate 5 ml of selective LB, which were grown overnight at 37 °C. The next morning, 3 ml of selective EZ containing 0.85 g l$^{-1}$ Pluronic F-127 (Sigma) were inoculated with the overnight preculture in a 1:100 ratio and grown for 3-4 h at 37 °C. Cells were centrifuged for 10 min at 4000 rcf and resuspended in ~10 μl of the supernatant to obtain a dense suspension, which was loaded into the PDMS microfluidics device. Cells were grown in a continuous culture inside microfluidic chambers (dimensions: 1.2 μm × 12 μm × 60 μm, h x w x l, purchased from Wunderlichips)[51] for 2 days with a constant 0.5 ml h$^{-1}$ supply of fresh medium (selective EZ plus 0.85 g l$^{-1}$ Pluronic F-127) and removal of waste and excess of bacteria, powered by an AL-300 pump (World Precision Instruments). For GRNs 2-OS.1, 1-OS.1, and 1-OS.2 the overnight and the subsequent 1:100 incubations contained 0.2% Ara, while 0.0001% Ara (for 2-OS.1) or 0.2% Ara (for 1-OS.1 and 1-OS.2) were used in the medium for the microfluidic experiment. For 1-OS.3, which lacks the $P_{BAD}$ promoter, no Ara was used in any of the media. Imaging was performed using a Leica DMi8 microscope and a Leica DFC9000 GT camera controlled by the Leica Application Suite X 3.4.2.18368, with the following settings: Cerulean: Ex. 440 nm 10% 50 ms, Em. 457–483 nm; mCitrine: Ex. 510 nm 10% 50 ms, Em. 520–550 nm; mCherry: Ex. 550 nm 20% 200 ms, Em. 600–670 nm; sfGFP: Ex. 470 nm 30% 200 ms, Em. 507–543 nm; mKO2 and mKate2 (indistinguishable): Ex. 550 nm 30% 200 ms, Em. 520–550 nm. Images were acquired every 10 min with LAS X software, and analyzed using Fiji[99] for montage.

### Mathematical modeling, overview

We developed an ensemble of mechanistic mathematical models that capture in detail level the biology of the transcription/translation/fluorescent maturation of various 2-node and 3-node GRNs. A model can be automatically generated depending on the GRN we want to simulate. Below, we decompose the model into its two core structures and assumptions. Next, we continue with additional functions we utilize to account for cell growth in the experiments.

### Modeling the dynamics of sgRNAs

The production rate of the sgRNA(s) depends on having either an inducible promoter (by arabinose) or a constitutively expressed promoter. The orange node (Fig. 3) is always induced by arabinose; the other two nodes (green and blue) have a constitutively expressed promoter. We also considered in our model the leakage of sgRNA(s) by the inducible promoter. The sgRNA(s) can bind reversibly to the catalytically-dead mutant dCas9, which forms a complex sgRNA:dCas. In turn, the sgRNA:dCas complex can bind reversibly and inhibit, via its

target sequence on the promoter (DNA), the respective gene expression, forming a sgRNA:dCas:DNA complex. We assume that RNAases act on degrading the sgRNA(s), whereas the complexes are affected by the dilution rate, m (see below for details on dilution). These biomolecular processes are described as chemical reactions.

Two chemical reactions represent the induction by arabinose and the leakage that a specific promoter has

$$\overset{Arabinose}{\rightarrow} sgRNA_i$$

$$\overset{leakage}{\rightarrow} sgRNA_i$$

and one chemical reaction depicts the case of the constitutive promoter:

$$\rightarrow sgRNA_i,$$

where the number of sgRNAs that is available is indicated with i and j, taking values from one to three based on the three reporter protein-coding genes that exist in the GRNs. RNA degradation (denoted by the empty symbol) as well as complex formation with dCas and DNA are modeled as:

$$sgRNA_i \rightarrow \varnothing$$

$$dCas + sgRNA_i \leftrightarrow dCas : sgRNA_i$$

$$dCas : sgRNA_i + DNA_j \leftrightarrow dCas : sgRNA_i : DNA_j$$

$$dCas : sgRNA_i : DNA_j \rightarrow DNA_j.$$

The final reaction corresponds to the combination of dilution of the [dCas: sgRNAi: DNA j] complex with DNA replication. Since dCas is constitutively expressed, the total concentration of dCas is conserved. The total promoter concentration (DNAj) is also conserved. These assumptions lead to the algebraic equations (using square brackets to denote concentrations):

$$[dCas_{total}] = [dCas] + \sum_i [dCas : sgRNA_i] + \sum_{i,j} [dCas : sgRNA_i : DNA_j]$$

$$(1)$$

$$[DNA_{total,j}] = [DNA_j] + [dCas : sgRNA_i : DNA_j] \qquad (2)$$

By using mass action kinetics (except for inducible promoters and dilution, see below), we transform the above chemical reactions and assumptions into a system of ordinary differential equations (ODEs):

$$\frac{d[sg\dot{R}NA_i]}{dt} = f_{(Ara)} + b_{sgRNA_i} - d_{RNA} \cdot [sgRNA_i] - k_{f_{ds}} \cdot [dCas] \cdot [sgRNA_i] + k_{r_{ds}} \cdot [dCas : sgRNA_i] \qquad (3)$$

$$\frac{d[sg\dot{R}NA_i]}{dt} = k_{sgRNA_i} \cdot [DNA_j] - d_{RNA} \cdot [sgRNA_i] - k_{f_{ds}} \cdot [dCas] \cdot [sgRNA_i] + k_{r_{ds}} \cdot [dCas : sgRNA_i] \qquad (4)$$

$$\frac{d[dCas:sgRNA_i]}{dt} = k_{f_{ds}} \cdot [dCas] \cdot [sgRNA_i] + k_{r_{dsd}} \cdot [dCas:sgRNA_i:DNA_j]$$
$$- m \cdot [dCas:sgRNA_i] - k_{r_{ds}} \cdot [dCas:sgRNA_i]$$
$$- k_{f_{dsd}} \cdot [dCas:sgRNA_i] \cdot [DNA_j] \tag{5}$$

$$\frac{d[dCas:sgRNA_i:DNA_j]}{dt} = k_{f_{dsd}} \cdot [dCas:sgRNA_i] \cdot [DNA_j]$$
$$- k_{r_{dsd}} \cdot [dCas:sgRNA_i:DNA_j]$$
$$- m \cdot [dCas:sgRNA_i:DNA_j] \tag{6}$$

Depending on whether the sgRNA is expressed by an inducible or a constitutive promoter, we use either the first or the second ODE. If there is an inducible promoter, then promoter leakage is constant, $b_{sgRNA_i}$. Production and degradation kinetic constants are $k_{sgRNA_i}$ and $d_{RNA}$; the binding and unbinding constants of sgRNA-dCas and sgRNA-dCas-DNA complexes are $k_{f_{ds}}$, $k_{r_{ds}}$ and $k_{f_{dsd}}$, $k_{r_{dsd}}$, respectively.

In addition, we assume a Hill function for promoter induction by arabinose (*Ara*):

$$f_{(Ara_{sgRNA_i})} = \frac{Ara^n}{Ara^n + K_m^{\;n}} \cdot k_2 \cdot [DNA_{total,j}] \cdot (1 - b_{sgRNA_i}) \tag{7}$$

where n is the Hill coefficient, $K_m$ is the affinity constant and $k_2$ is the production constant.

## Dynamics of reporter protein-coding genes

Similar to the production rate of sgRNAs, the mRNA production of the three reporters used in this study depends either on the activity of an inducible promoter or on a constitutive promoter (expression of the orange node via arabinose induction, green and blue node via constitutive promoter). mRNA translation produces protein $PI_j$, which matures to the final fluorescent reporter ($P_j$). mRNA is degraded at the same rate as the degradation of sgRNAs. However, the reporter proteins (immature and fluorescent) undergo active degradation (which depends on the reporter) and dilution (see growth model section). The chemical reactions that summarize the expression of the reporter proteins are:

$$\xrightarrow{Arabinose} mRNA_j$$

$$\xrightarrow{leakage} mRNA_j$$

$$\rightarrow mRNA_j$$

$$mRNA_j \rightarrow \varnothing$$

$$mRNA_j \rightarrow mRNA_j + PI_j$$

$$PI_j \rightarrow \varnothing$$

$$PI_j \rightarrow P_j$$

$$P_j \rightarrow \varnothing \, .$$

Again, we differentiate between arabinose-inducible and constitutive promoters and account for leakage. By assuming mass-action

kinetics, the ODE system is:

$$\frac{d[\dot{mRNA_j}]}{dt} = f_{(Ara_{mRNA_j})} + b_{mRNA_j} - d_{RNA} \cdot [mRNA_j] \tag{8}$$

$$\frac{d[\dot{mRNA_j}]}{dt} = k_{mRNA_j} \cdot [DNA_j] - d_{RNA} \cdot [mRNA_i] \tag{9}$$

$$\frac{d[\dot{PI_j}]}{dt} = k_{PI_j} \cdot [mRNA_j] - m_{PI_j} \cdot [P_j] - (m + d_{PI_j}) \cdot [PI_j] \tag{10}$$

$$\frac{d[\dot{P_j}]}{dt} = m_{PI_j} \cdot [PI_j] - (m + d_{PI_j}) \cdot [P_j] \tag{11}$$

Kinetic constants of production, degradation, and promoter leakage are $k_{mRNA_j}$, $d_{RNA}$ and $b_{mRNA_j}$. Translation, maturation and active degradation constants are $k_{PI_j}$, $m_{PI_j}$ and $d_{PI_j}$ respectively. In addition, $f_{(Ara_{mRNA_j})}$ is the same function as above, but for $mRNA_j$:

$$f_{(Ara_{mRNA_j})} = \frac{Ara^n}{Ara^n + K_m^{\;n}} \cdot k_2 \cdot [DNA_{total,j}] \cdot (1 - b_{mRNA_j}) \tag{12}$$

## Growth model

We represent the cell population growth during the microplate experiment, *G(t)*, by a superposition of three generalized logistic functions. This approach allows capturing events such as diauxic shifts in which growth slows down due to limited nutrient availability. Specifically:

$$G(t) = p_1 + \sum_{k=1}^{3} \frac{p_2^k}{(1 + p_3^k e^{-t \cdot p_4^k})^{1/p_5^k}} \tag{13}$$

In our model, the dilution rate *m* is the ratio of the derivative of *G(t)* and G(t) – that is, the specific growth rate – with parameters estimated from the appropriate data (see Data S1). The rest of the experimental data was used for independent model validation.

## Initial conditions

Prior to the microplate experiments, the uninduced cells where precultured in Luria Broth medium for approximately six hours. We capture these initial conditions using the growth data from[100] and simulating the appropriate model for six hours.

## Model parametrization

Depending on the total number of inhibitions, the ODE models differ in the mRNA degradation constants and the total concentration of dCas (e.g. all the 3-node models with three inhibitions have a different mRNA degradation compared to the 3-node models with six inhibitions). In addition, the translation constant of the orange node is arabinose dependent (see details below). Because our plasmids are of medium copy number (i.e. 25-30 copies), and assuming that one copy is 1nM[20], we used a total concentration of all promoters (DNA_A, DNA_B, DNA_C) of 30 nM. Finally, we assumed that an optical density at 600 nm of one corresponds to $8 \cdot 10^8$ E. coli cells.

To estimate the remaining model parameters, initially, we turned to the literature[101,102] to obtain the biologically relevant ranges (e.g., mRNA degradation, binding or unbinding of dCas complexes). Next, parameters were estimated by using the enhanced scatter search method[103], a robust method for solving non-linear global optimization

problems. We defined an objective according to:

$$\sum_{i=1}^{N}\left(\frac{x_i - x_i^s(\theta)}{\sigma_i}\right)^2 \xrightarrow{\theta} \min \tag{14}$$

that is, the minimization of the weighted squared residuals. Specifically, N is the number of data points used for parameter estimation, $x_i$ is the vector of experimental data, $\sigma_i$ is the corresponding measurement variance (for details, see below and Supplementary Fig. 5e) and $x_i^s$ are the relevant state variables of the ODE models. To evaluate a model fit, we used the $\chi^2$-test, with $\chi^2 = 27099,71$ estimated parameters and 27312 degrees of freedom.

Given that all models share most of their parameters, we performed the estimation simultaneously for all experimental data and models used. Specifically, we used all the 2-node experimental data (9 different arabinose inputs, 6 different inhibition strengths, 100 time points) and four 3-node GRNs (1.4, 2cNF.1, 3.2, 4.1). The rest of the experimental data was used for independent model validation.

## Promoter efficiencies

The $P_H$ (BBa J23100) and $P_M$ (BBa J23102) promoters that were used for building some of the synthetic GRNs are from a standard collection recovered from a library screen[104]. Therefore, in our modeling process, we used the measured strength 1 and 0.86, respectively. Concerning the $P_L$ (BBa J23150) promoter, we found discrepancies between previously determined promoter strengths[105,106] and our estimates, possibly due to context-dependent effects on gene expression[107]. Specifically, we identified common promoters (BBA J23102, BBA J23116, BBA J23113) in the dataset from Kelly et al. and the Anderson collection. Using linear regression, we predicted the promoter efficiency of the $P_L$ we would have observed if it had been measured in the Anderson collection. With the same procedure, we compared the Davis dataset and the Anderson collection (common promoters: BBA J23101, BBA J23113). For the two comparisons, we predicted for the $P_L$ promoter a relative strength of 0.3 and 0.2, respectively. However, from our estimation, we predicted that the strength was 0.75. Given that we have the additional control analysis which contains the relative strengths identified by[105,106], we continued with our estimated strength when comparing to the experimental data.

## Translation of mRNA induced by arabinose

The expression of the orange node (reporter protein: mKO2) is induced by arabinose, controlling the expression of the mKO2 mRNA, which then translates into the immature protein of mKO2. When calibrating the mathematical models we noticed non-linearities in the arabinose response, although we used the MK01 strain[97]. A possible explanation[108] is the difference in the number of transporters per cell in the population; cells can accumulate sufficient arabinose (or not) to overcome the internal threshold concentration for gene activation. We tackled this observation phenomenologically by making the translation constant dependent on the concentration of arabinose, independent of the modeled topologies.

## Scaling parameters and measurement models

To map the model output ($P_A$, $P_B$, $P_C$) to the microplate reader experiments, we included three scaling factors (for the three fluorescent reporters) in the optimization procedure (Data S1). Given that the experiment for each reporter was performed with the same settings, we assumed three measurement models. We performed linear regression on the variance (s.d.) as a function of average fluorescent signal per reporter. We used the following models per reporter, with its

average fluorescent signal x (Supplementary Fig. 5):

$$GFP(x) = \begin{cases} 10350, & x < 7e+04 \\ 0.13 \cdot x + 1250, & 7e+04 \leq x \leq 2.5e+05 \\ 3.375e+04, & x > 2.5e+05 \end{cases} \tag{15}$$

$$mKO2(x) = \begin{cases} 899, & x < 600 \\ 0.08 \cdot x + 419, & 6e+03 \leq x \leq 2e+04 \\ 2e+03, & x > 2e+04 \end{cases} \tag{16}$$

$$mKate2(x) = \begin{cases} 43, & x < 500 \\ 0.03 \cdot x + 23, & x \geq 500 \end{cases} \tag{17}$$

## Definition of a stripe

To characterize a GRN as a stripe-forming one, initially, we normalized the simulated data. We divided each reporter model output ($P_A$, $P_B$, $P_C$) at 900 min for the different arabinose concentrations by the maximum observed for each reporter (at 900 min for the same arabinose concentrations). Next, we identified the node that reached the maximum peak (i.e. a value of 1) when the other two were at the minimum. To accept or reject a stripe (even if a node was identified), we set a threshold of 6% increase and decrease, concerning the average of two lower and two higher arabinose induction levels with respect to the maximum point. Therefore, a functional (BLUE- or GREEN-stripe) phenotype should on average at the two lowest and two highest arabinose concentrations have at least an increase of 6% and a decrease of 6%. We designated all other phenotypes as non-functional.

## Mutational change in GRNs

Within the modeling framework, we represent each GRN as an 8-length vector. The first six elements of the vector represent the positions of an inhibition, similar to Fig. 4e (or absence, represented with 0), and the last two elements represent the type of promoter for the blue and green nodes. In the genotype networks, we define one mutational change as the 1-Hamming distance between any one change in the 8-length vectors (i.e. 1-neighborhood).

## Genotype maps of synthetic GRNs

We created genotype maps by finding all the possible combinations of the modular parts at our disposal (i.e. qualitative and quantitative changes) and evaluating them for observing a non-functional or functional phenotype. A single set of parameters (that best minimizes the objective function) was used for all the forward simulations. Specifically, we had: (1) four different promoter efficiencies (the $P_L$, $P_M$, $P_H$ and the 0 case) with two available promoter positions (green and blue node), (2) six sgRNAs, and (3) 42 topologies with a minimum of three and a maximum of six inhibitions. We did not include topologies with only one or two inhibitions due to the expected reduced functionality these topologies[33,60]. Therefore, for each topology, the total number of GRNs is:

$$T_{(E)} = p^2 \cdot s^E \tag{18}$$

where p is the total number of promoter efficiencies, s is the total number of sgRNAs, and E depends on the number of total inhibitions a specific topology has. For the control analysis with artificial parameter values (a regular grid in parameter space), the number of modular parts consisted of: (1) five evenly spaced (from 0 to 1) promoter efficiencies with two available promoter positions (green and blue node), (2) five sgRNAs with evenly spaced kinetic dissociation constants (from 5 to 45 nM), and (3) 42 topologies.

## Genotype maps of oscillatory synthetic GRNs

To generate the genotype maps of oscillatory synthetic circuits, we used the same mathematical models and parametrization as for the stripe-forming GRNs. However, we assumed that the binding of the species for the sgRNA-dCas-DNA complex formation is irreversible (similarly assumed here)[109]. This means that the kinetic constant $k_{r_{dsd}}$ was set to zero. Given the nature of the microfluidic experiment, we assumed constant growth of 1e-03 min$^{-1}$ and an effective arabinose concentration of 1e-05 %. In addition, we used the parameter values of sgRNAs that were experimentally evaluated (six sgRNAs: sgRNA1, sgRNA1t4, sgRNA2, sgRNA3, sgRNA4, sgRNA4t4 and four promoter efficiencies: 1, 0.86, 0.75, 0). To automatically detect whether a GRN is oscillating or not, we used the normalized autocorrelation method (at zero lag the sequence is identical) with a peak detection of at least 0.01[110,111].

## Connectivity of genotype networks

A connected component of an undirected graph is the maximal set of nodes such that each pair of nodes is connected by a path[112]. To evaluate the connectivity for the genotype networks, we utilized the command *components* from the *igraph* package[113] in Rstudio[114]. It uses either a simple breadth-first search or two consecutive depth-first searches.

## Robustness per perturbation

As in Catalán et al.[62], we measured the fraction of neighbors that maintain the same phenotype per perturbation (i.e. changes in sgRNAs or in promoter efficiencies), termed here as robustness per perturbation (or neutrality, see Catalán et al.[62]). We randomly sampled 1% of GRNs per functional phenotypes (2000 GRNs) and 0.1% of non-functional phenotype (2000 GRNs) and evaluated all their neighbors (which accounts for ~4.5% of the total possible GRNs). We tested various sampling schemes, until convergence was observed (Supplementary Tables 4–6).

## Random walks

To determine the path length for alternating phenotypes (GREEN-stripe, BLUE-stripe, and NF phenotypes), we randomly selected 5% GRNs per functional phenotype and 2% for the non-functional one (~10700 GRNs with BLUE-stripe and GREEN-stripe, and 28903 GRNs for NF). From these starting GRNs, the algorithm proceeded by uniform random selection of a GRN from the set of neighboring GRNs. We stopped the walk upon a change in phenotype or when the maximum number of steps (50 for functional phenotypes, 300 for NF) was reached. We used the built-in command *random_walk* from the *igraph* package[113] in RStudio[114].

## Prevalence of epistasis

To measure the occurrence of epistasis within our genotype networks we sampled again 5% GRNs per functional phenotype and 2% for the non-functional one. Our goal was for each sampled GRN to evaluate if it is part of an orthogonal network of GRNs, in which only one of the direct neighbors will differ in the phenotype. We did this by evaluating the 1- and 2-Hamming distance neighborhoods of the sampled GRN.

## Reporting summary

Further information on research design is available in the Nature Portfolio Reporting Summary linked to this article.

## Data availability

The source data and code for generating the model-related figures (Figs. 3b, 3d, 4a–e, 5a, 5b, 6b, 7b, 7c and Supplementary Figs. 2–5) are available at https://doi.org/10.3929/ethz-b-000604092. Model parameters are provided in Supplementary Data 1. The plasmids used in this study are listed in Supplementary Table 1 and their annotated sequences are provided (Supplementary Data 2). The source data underlying Figs. 2, 3b, 3c, 4f, 6c, and 7a, and Supplementary Fig. 1 are provided as a Source Data file (Source data). A comparison of promoter efficiencies was done using data available in Anderson et al.[104], Davis et al.[105], and Kelly et al.[106]. Source data are provided with this paper.

## Code availability

Custom code for the analysis and generation of model-related figures is available at https://doi.org/10.3929/ethz-b-000604092.

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

## Acknowledgements

We thank Clement Gallay and Jan-Willem Veening for help with microscopy, Ruben Pérez-Carrasco and Björn Vessman for exploratory

CRISPRi stripe models, and Joshua Payne and Berta Verd for useful feedback on a previous version of the manuscript. We also thank Mariapia Chindamo, Aysun El Wardani, Florence Gauye, Virginie Kahabdian, Borany Kim, Léo Moser, Antoine Triani and Lucie Gilliéron for help with cloning, and all Schaerli lab members for useful discussions. This work was funded in part by the Swiss National Science Foundation (grants 31003A_175608 and 310030_200532 awarded to Y.S).

## Author contributions

J.S.M. and Y.S. designed the experimental research. J.S.M. and H.K. performed experiments and analyzed data. E.T. and J.S. designed the computational part and performed the mathematical modeling. J.S.M., Y.S., E.T., and J.S. wrote the manuscript. All authors revised the manuscript and approved the final version of the manuscript.

## Competing interests

The authors declare no competing interests.
