## [Peer Review File · Nature Communications]

Reviewers' Comments:

Reviewer #1:

Remarks to the Author:

This is a very impressive piece of research carried out by Santos-Moreno et al., continuing their previous work from 2020, also published in Nat. Comms. The authors set out to design a large number of synthetic three-node gene regulatory networks, and study the corresponding genotype-phenotype map both experimentally and computationally, through a mathematical model of RNA and protein expression dynamics.

This work is very good and relevant, connecting the world of computational genotype-phenotype maps with synthetic biology and experimental studies. I support its publication in this journal. However, I have some minor questions and comments, see below:

COMMENTS

1) L229: You consider 42 topologies with 3 to 6 edges? Why did you restrict yourselves to this number? Why not consider topologies with 1 or 2 edges? Is it because of reduced functionality?

2) In line 649, you mention this reaction: $dCas : sgRNAi : DNA_j \rightarrow DNA_j$
What does this stand for? I don't see any term in the ODEs reflecting this.

3) The equations in lines 658 and 659 should include the terms $-k_{f_ds} [dCas] [sgRNAi]$ and $k_{r_ds} [dCas : sgRNAi]$, as in the Supplementary Material. I understand the equations are correctly implemented in MATLAB.

4) In line 706, you state that the dilution rate is $G'(t)$, but shouldn't it be $G'(t)/G(t)$? I
For instance, in standard systems biology models that assume exponential growth, the dilution rate is a constant k , as $G'(t)=kG(t)$.
This is especially relevant for the initial phases of growth, where $G(t)$ is quite low and, without the normalization factor, you'd get a very low dilution rate, while in fact it is quite large! (as cells are growing almost exponentially)

5) While the model works quite well in general, it's interesting that it fails to capture the behavior of the blue reporter at low Ara concentrations (Fig. S2 or Fig. 3C). Is it because of point 4 raised before, or are there other aspects that the model might be missing? Aside from the error I spotted, I think the model is quite good and complete, so I'm curious if you have any intuition of what might be missing.

6) In line 719, you say "all the 3-node models with three inhibitions have a different mRNA degradation compared to the 3-node models with six inhibitions". Why is this? Is it because the concentration of RNAses is constant? Couldn't you include this in the model with an additional set of reactions? It seems this would reduce the number of needed parameters, right? (It's fine if you don't want to do this, I'm just curious)

7) I'm not sure I understand the definition of a stripe (L784). Would a green M-shape (where green expression first increases, then decreases, then increases again and decreases once more) be considered a green stripe? Or is it the case that you don't get phenotypes like these?

8) In line 799, you say: "we represent each GRN as an 8-length vector. The first six elements...". In Figure 4 it becomes clear that the last two components are promoter strengths, but this is not stated in the Methods and should be done.

9) You measure robustness by sampling, but actually you could get it exactly, as the network is not too big ($\sim 2e6$ nodes is feasible in my experience). If you save a file assigning each genotype to each phenotype (GREEN, BLUE or NF), then it's easy to write a breadth-first (or depth-first) search algorithm that gives you the exact robustness for each node. In fact, given the size of your network, you could even do it with a brute-force for loop, going through every node and their neighbors. If you feel this is too much work, you don't have to do it, as I expect the results will not

differ much from the samples you've obtained.

10) in line 355 you say "This is consistent with a previously observed trade-off between genotypic evolvability and robustness (Payne et al., 2014), but apparently contradicts the finding that complexity aids robustness (Catalán et al., 2018)"

Your results are indeed in agreement with that of Payne et al., because genotypic evolvability and robustness cannot be positively correlated: higher robustness means less available mutants to other phenotypes, and therefore a lower boundary for evolvability.

However, I am not sure these results contradict Catalán et al.: their results suggest that adding more complexity to a GP map would result in both increased robustness and evolvability. In the framework of this paper, Catalán et al.'s thesis would imply adding more genes to the GRN. If their results are correct, then we would expect four-node GRNs expressing a GREEN phenotype, for instance, to be more robust and evolvable than their three-gene counterparts. But this is not something you have tested here (Although it would be very interesting!)

Reviewer #2:

Remarks to the Author:

The manuscript presents work exploring the robustness and evolvability of gene regulatory networks. The authors take advantage of a recently developed synthetic gene regulatory network using CRISPR guided repression of three genes encoding three different fluorescent proteins in *E. coli*. This synthetic network originally produced a peak of GFP expression at a specific concentration of Arabinose. When Arabinose is spatially variable in concentration, this is observable as a "stripe" of green cells between red and orange cells. In the current manuscript, the authors experimentally manipulated this network by varying the promoter strength of each of the three proteins, and by adding and deleting gRNAs to change the topology of the regulatory network. Following this experimental analysis, they further developed an ODE model that recapitulates some of the data, and then more thoroughly explore how changes to the regulatory network change the gene expression output using the model. By considering a change in one parameter of the model to be a "mutation" they build and analyze genotype networks where a parameter change keeps the same stripe phenotype. To understand evolvability, they analyze how many of the individual parameter changes result in a new phenotype of either a different colored stripe, or no stripe (termed non-functional). The results are interesting, and provide a novel glimpse at the evolutionary potential of regulatory networks that result in spatial (and temporal) patterns of gene expression, which are of fundamental importance for developmental biology. I am in favor of publication. The following recommendations are provided to improve the readability of the manuscript.

Suggestions for authors:

The title is vague and should be improved to be more descriptive.

The abstract should make some mention of the approach, i.e. that gene expression networks are made using CRISPR guided gene repression of three genes that encode different fluorescent proteins, which is measured by spatially resolved fluorescence, or equivalent.

Line 50 "comparative studies" could be briefly explained. Because of the expected broad audience that this paper would appeal to, it is unlikely that all of them would be familiar with these references. One paper compared *C. elegans* to other nematodes. Do all of them involve model organisms compared to closely related wild species?

The authors should consider their analysis in terms of environmental change? A brief acknowledgment of the importance of environmental dependence of phenotypes with references should be added to the introduction (~line 73). If the authors have considered how their model system can inform this relationship, this could be added to the Discussion.

The manuscript needs to describe why this regulatory pattern is referred to as a "stripe". In the original manuscript, a stripe was observed spatially when an arabinose source is placed in the

center of a dish. This is important because it makes the connection to developmental biology more clear. But, this original experimental observation needs to be clearly explained in the introduction because this same spatial pattern is not presented in the main text. According to the Methods, the experiments were done by putting different concentrations of arabinose in different wells of a plate. This approach is fine, but the intro needs to clarify the stripe terminology if it is going to be used. If possible, image of the stripes could be added to figure 1a? As written it actually sounds like the stripe might be inside each individual bacterium.

Near Line 113, add reference to the literature that developed the CRISPRi approach

This discussion of Fig. 4E is hard to follow. Please clearly define "robustness" in this context. Is only green to blue transition considered not robust? Why not green to NF? Please clarify in the main text.

Line 427, The conclusion that stripe genotypes facilitate innovation to oscillators could use some further analysis. Is there an estimate of how many functional stripe GRNs can evolve to oscillate? It would seem that many of the "3.x" networks could be converted to oscillators with a single change, for example. Is the repression going blue to green changing to green to blue considered a single mutation, or two? My logic is that the network only facilitates innovation if some, but not all nodes, can mutate to oscillators. This seems likely, but is not addressed.

Minor suggestions:

I very much appreciated the discussion of the size of mutational changes. However, I was thinking about this the whole time I was reading which was distracting. Could some of this be added to the introduction (near line 86), or the results paragraphs?

On my version, the font in Fig 1A is too small to read the promoter subscripts. This is a problem because it is needed to fully evaluate the logic of the network.

Fig. 2, I recommend labelling the GREEN-stripe and BLUE-stripe networks on the figure, not just in the legend. Some text to the right or left of the group should work.

Line 282, should this read ". . .either BLUE- or GREEN-stripe phenotypes"? instead of "both" "and"

Line 353, clarify what "substantially better" means

Line 418, a description of the logic of the oscillator would be nice. Why does it oscillate?

Line 520, "hamper" is a poor word choice. How does the analysis of epistasis hamper our understanding? I would think the analysis promotes our understanding.

Robustness and innovation in synthetic genotype networks

Javier Santos-Moreno*, Eve Tasiudi*, Hadiastri Kusumawardhani, Joerg Stelling#, Yolanda Schaerli#

Point-by-point response to reviewers' comments

Reviewer #1 (Remarks to the Author):

This is a very impressive piece of research carried out by Santos-Moreno et al., continuing their previous work from 2020, also published in Nat. Comms. The authors set out to design a large number of synthetic three-node gene regulatory networks, and study the corresponding genotype-phenotype map both experimentally and computationally, through a mathematical model of RNA and protein expression dynamics.

This work is very good and relevant, connecting the world of computational genotype-phenotype maps with synthetic biology and experimental studies. I support its publication in this journal. However, I have some minor questions and comments, see below:

We thank the reviewer for the encouraging words and the constructive feedback. Reviewer's comments have been addressed and discussed below:

COMMENTS

1) L229: You consider 42 topologies with 3 to 6 edges? Why did you restrict yourselves to this number? Why not consider topologies with 1 or 2 edges? Is it because of reduced functionality?

For this study, we solely focused on networks that could yield a stripe phenotype. Given that we only had inhibitory (CRISPRi) interactions between the nodes and considering previous studies (Schaerli et al., Nat Commun 5, 4905 (2014); Cotterell & Sharpe, Molecular Systems Biology 6, 425 (2010)) that indicate the requirement of a feed-forward architecture for this phenotype, we decided to only look into three-node networks with a minimum of 3 edges. We have now included the following clarification in the Methods part in the section of *Genotype maps of synthetic GRNs* (line 813): “We did not include topologies with only one or two inhibitions due to the expected reduced functionality these topologies^{33,60}.”

2) In line 649, you mention this reaction: $dCas : sgRNAi : DNA j \rightarrow DNA j$ What does this stand for? I don't see any term in the ODEs reflecting this.

This reaction corresponds to the dilution of the complex $[dCas : sgRNAi : DNA j]$. However, because dilution is balanced by replication for the DNA (i.e., the promoter concentrations should be constant), we capture the combination of the two processes with this reaction. In the system of ODEs it is specified by the last factor in the equation of the species $[dCas : sgRNAi : DNA j]$. For clarification, we have added in line 651 the following: “The final reaction corresponds to the combination of dilution of the $[dCas : sgRNAi : DNAj]$ complex and DNA replication.”

3) The equations in lines 658 and 659 should include the terms $-k_{f_ds} [dCas] [sgRNAi]$ and $k_{r_ds} [dCas : sgRNAi]$, as in the Supplementary Material. I understand the equations are correctly implemented in MATLAB.

Thank you for spotting the typographical error. In the Supplementary Material and in the MATLAB implementation, these terms are indeed correctly annotated. We have added the missing terms in the respective equations (lines 660-661).

4) In line 706, you state that the dilution rate is $G'(t)$, but shouldn't it be $G'(t)/G(t)$? I For instance, in standard systems biology models that assume exponential growth, the dilution rate is a constant k , as $G'(t) = kG(t)$. This is especially relevant for the initial phases of growth, where $G(t)$ is quite low and, without the normalization factor, you'd get a very low dilution rate, while in fact it is quite large! (as cells are growing almost exponentially)

Thank you for spotting this error – indeed, the specific and not the absolute growth rate is required for the models. In addition to correcting the text (Methods, line 708), we have re-estimated parameters for the modified models, and re-computed all predictions and analyses. Main text figures 3-6, the corresponding data in the main text, as well as corresponding Supplementary figures and tables have been updated with the new data. The differences to our previous results are only minor, we assume, because the stripe phenotypes evaluated at late time points are not very sensitive to the initial growth dynamics. Importantly, these updates do not affect the conclusions about the data discussed in the text (for example, the most visually apparent difference is in slight re-ordering of topologies in Fig. 5b).

5) While the model works quite well in general, it's interesting that it fails to capture the behavior of the blue reporter at low Ara concentrations (Fig. S2 or Fig. 3C). Is it because of point 4 raised before, or are there other aspects that the model might be missing? Aside from the error I spotted, I think the model is quite good and complete, so I'm curious if you have any intuition of what might be missing.

For parameter estimation, we used the entire time-course data generated from microplate reader experiments, which means that fits and predictions for the stripe phenotype are not necessarily optimal. Specifically, to evaluate whether a specific circuit can generate a stripe, we normalized the fluorescence values at 900min for the different arabinose induction concentrations with the maximum value observed at that specific time point for each individual fluorescence color. We have done this both for the experimental and the simulated data. However, this type of normalization has the effect that differences among larger fluorescence values are condensed, whereas differences between smaller values are enlarged (please see Figure below for the effect of changing the normalization scheme to a min-max normalization (Fig 1b)). For the arabinose concentrations where the fluorescent proteins show a sigmoidal increase or decrease in their concentration the effect is not pronounced. This is, we believe, the main reason why we see this inconsistency between the simulated stripes and the experimental stripes. However, to be consistent with our previous publication (Santos-Moreno et al., Nat Commun 11, 2746 (2020)) we analyzed the data the same way.

Figure 1: a) Trajectories with experimental data in blue and model (black solid line) for the 2c2 circuit for the blue node (mKate2 fluorescent protein) for four different arabinose induction levels. b) Circuits that were used in the optimization process (shown in Supplementary Fig. 2) with changed normalization procedure.

6) In line 719, you say “all the 3-node models with three inhibitions have a different mRNA degradation compared to the 3-node models with six inhibitions”. Why is this? Is it because the concentration of RNAses is constant? Couldn't you include this in the model with an additional set of reactions? It seems this would reduce the number of needed parameters, right? (It's fine if you don't want to do this, I'm just curious)

Indeed, the biological assumption is that with increasing numbers of sgRNA, RNase activity can become limiting (which we concluded from simulation results indicating this phenomenon, and the effect of introducing different mRNA degradation constants, which substantially improved the model fits). In principle, we could include a more mechanistic representation in the model via extra reactions as suggested, but this would not change the number of required parameters substantially (we would need at least association, dissociation, and catalytic rate constant parameters for the RNase; our phenomenological model now employs four parameters for the 3-6 inhibitory interactions). We therefore did not change the model.

7) I'm not sure I understand the definition of a stripe (L784). Would a green M-shape (where green expression first increases, then decreases, then increases again and decreases once more) be considered a green stripe? Or is it the case that you don't get phenotypes like these?

We did not observe phenotypes like the M-shape in our data, and therefore used the simple definition for a stripe (maximum and constraint on neighboring increase / decrease). However, one can apply this definition multiple times per fluorescence profile, and the M-shape would then be classified as functional (with two stripes, if both reach maximum fluorescence, and else with a single stripe).

8) In line 799, you say: “we represent each GRN as an 8-length vector. The first six elements...”. In Figure 4 it becomes clear that the last two components are promoter strengths, but this is not stated in the Methods and should be done.

We have now clarified this point in the Methods section, line 799: ‘The first six elements of the vector represent the positions of an inhibition, similar to Fig. 4e (or absence, represented with 0), and the last two elements represent the type of promoter for the BLUE and GREEN nodes.’

9) You measure robustness by sampling, but actually you could get it exactly, as the network is not too big (~2e6 nodes is feasible in my experience). If you save a file assigning each genotype to each phenotype (GREEN, BLUE or NF), then it's easy to write a breadth-first (or depth-first) search algorithm that gives you the exact robustness for each node. In fact, given the size of your network, you could even do it with a brute-force for loop, going through every node and their neighbors. If you feel this is too much work, you don't have to do it, as I expect the results will not differ much from the samples you've obtained.

Thank you for the suggestion. For calculating the robustness per perturbation per each phenotype, we started with sampling 500 nodes, 2000 nodes and finally 20000 nodes. As shown in the new supplementary tables 4, 5 and 6 for the individual phenotypes, we did not see any difference between the mean (and standard deviation) for each change in the gene regulatory network. As expected, and supported by these sampling data, the robustness calculations are very stable, such that we did not perform additional exhaustive enumeration.

10) in line 355 you say “This is consistent with a previously observed trade-off between genotypic evolvability and robustness (Payne et al., 2014), but apparently contradicts the finding that complexity aids robustness (Catalán et al., 2018)”

Your results are indeed in agreement with that of Payne et al., because genotypic evolvability and robustness cannot be positively correlated: higher robustness means less available mutants to other phenotypes, and therefore a lower boundary for evolvability.

However, I am not sure these results contradict Catalán et al.: their results suggest that adding more complexity to a GP map would result in both increased robustness and evolvability. In the framework of this paper, Catalán et al.’s thesis would imply adding more genes to the GRN. If their results are correct, then we would expect four-node GRNs expressing a GREEN phenotype, for instance, to be more robust and evolvable than their three-gene counterparts. But this is not something you have tested here (Although it would be very interesting!)

Thank you for this clarification; we have now deleted the second half of the sentence referring to the work by Catalan et al. (which measured complexity by gene and not interaction number as we did). We agree that such an analysis would be very interesting, and we have added a corresponding statement to the Discussion (line 492): “Future work could extend this approach to investigate the effect of varying gene numbers on robustness and evolvability, as in prior work with abstract GRNs⁶⁴.”

Reviewer #2 (Remarks to the Author):

*The manuscript presents work exploring the robustness and evolvability of gene regulatory networks. The authors take advantage of a recently developed synthetic gene regulatory network using CAS9 guided repression of three genes encoding three different fluorescent proteins in *E. coli*. This synthetic network originally produced a peak of GFP expression at a specific concentration of Arabinose. When Arabinose is spatially variable in concentration, this is observable as a “stripe” of green cells between red and orange cells. In the current manuscript, the authors experimentally manipulated this network by varying the promoter strength of each of the three proteins, and by adding and deleting gRNAs to change the topology of the regulatory network. Following this experimental analysis, they further developed an ODE model that recapitulates some of the data, and then more thoroughly explore how changes to the regulatory network change the gene expression output using the model. By considering a change in one parameter of the model to be a “mutation” they build and analyze genotype networks where a parameter change keeps the same stripe phenotype. To understand evolvability, they analyze how many of the individual parameter changes result in a new phenotype of either a different colored stripe, or no stripe (termed non-functional). The results are interesting, and provide a novel glimpse at the evolutionary potential of regulatory networks that result in spatial (and temporal) patterns of gene expression, which are of fundamental importance for developmental biology. I am in favor of publication. The following recommendations are provided to improve the readability of the manuscript.*

We thank the reviewer for the constructive feedback. We addressed the comments and think that these changes improved the manuscript. We hope the reviewer agrees with us.

Suggestions for authors:

The title is vague and should be improved to be more descriptive.

We have now changed the title to “Robustness and innovation in synthetic genotype networks”. We did not want to choose a title that would contain the word “network twice, such as “Genotype network of synthetic gene regulatory networks.”

The abstract should make some mention of the approach, i.e. that gene expression networks are made using CRISPR guided gene repression of three genes that encode different fluorescent proteins, which is measured by spatially resolved fluorescence, or equivalent.

The abstract now mentions the approach: “Our synthetic GRNs contain three nodes regulating each other by CRISPR interference and governing the expression of fluorescent reporters.”

Line 50 “comparative studies” could be briefly explained. Because of the expected broad audience that this paper would appeal to, it is unlikely that all of them would be familiar with these references. One paper compared C. elegans to other nematodes. Do all of them involve model organisms compared to closely related wild species?

As suggested, comparative studies are now briefly explained. The sentence now reads “It mainly comes from comparative analyses of the expression dynamics and regulatory structure of GRNs in related species showing that rewiring of GRNs during the course of evolution does not necessarily alter the resulting gene expression pattern²²⁻²⁶”

The authors should consider their analysis in terms of environmental change? A brief acknowledgment of the importance of environmental dependence of phenotypes with references should be added in the introduction (~line 73). If the authors have considered how their model system can inform this relationship, this could be added to the Discussion.

In the introduction we mention now: “Furthermore, epistasis can itself be dependent on environmental conditions, such as the temperature, the medium, concentration of an expression inducer or an enzymatic co-factor⁴¹⁻⁴³.”

In this study, we did not investigate the environment-dependent epistasis. However, we used very similar synthetic circuits to systematically study environment-dependent epistasis and saw that it increases phenotypic diversity. If this is of interest, we invite the reviewer to check out our preprint: <https://www.biorxiv.org/content/10.1101/2022.09.18.508240v1> We now also cite this reference in the introduction.

The manuscript needs to describe why this regulatory pattern is referred to as a “stripe”. In the original manuscript, a stripe was observed spatially when an arabinose source is placed in the center of a dish. This is important because it makes the connection to developmental biology more clear. But, this original experimental observation needs to be clearly explained in the introduction because this same spatial pattern is not presented in the main text. According to the Methods, the experiments were done by putting different concentrations of arabinose in different wells of a plate. This approach is fine, but the intro needs to clarify the stripe terminology if is going to be used. If possible, image of the stripes could be added to figure 1a? As written it actually sounds like the stripe might be inside each individual bacterium.

As suggested, the stripe terminology has been clarified, and the Introduction now includes the following: “Our IFFL-2 has been shown to produce a low-high-low gene expression pattern (“stripe” pattern) across a bacterial population in response to a chemical concentration gradient

(Fig. 1a)⁵¹, analogous to the formation of spatiotemporal gene expression patterns guided by morphogen gradients during development^{54, 55}. In the present study, we characterized a large number of IFFL-2-derived GRNs by incubating bacteria at discrete concentrations of a chemical inducer and evaluating the expression pattern across the concentration range.”

Fig. 1a has also been modified to display a schematic of the stripe pattern across a bacterial population.

Near Line 113, add reference to the literature that developed the CRISPRi approach

A reference to Qi et al. 2013 has been added.

This discussion of Fig. 4E is hard to follow. Please clearly define "robustness" in this context. Is only green to blue transition considered not robust? Why not green to NF? Please clarify in the main text.

We have added the following context-specific definition in the main text (line 313): “As in prior work⁶², we define robustness of a phenotype (GREEN-stripe, BLUE-stripe, or non-functional) of a reference GRN by the fraction of neighboring GRNs that have the same phenotype when we apply a single mutation (i.e., changing an inhibitory interaction or a promoter strength).” In other words, any transition away from the reference GRN’s phenotype is considered not robust, represented in Fig. 4d as off-diagonal elements. We performed the analysis for reference BLUE-stripe GRNs (Fig. 4e), GREEN-stripe GRNs and NF GRNs (Supplementary Fig. 3), which the updated text now clarifies.

Line 427, The conclusion that stripe genotypes facilitate innovation to oscillators could use some further analysis. Is there an estimate of how many functional stripe GRNs can evolve to oscillate? It would seem that many of the “3.x” networks could be converted to oscillators with a single change, for example. Is the repression going blue to green changing to green to blue considered a single mutation, or two? My logic is that the network only facilitates innovation if some, but not all nodes, can mutate to oscillators. This seems likely, but is not addressed.

Thank you for the suggestion – we have now included a systematic analysis of oscillator phenotypes and their robustness analogous to the one for stripe-forming GRNs (see new paragraph in section ‘A genotype network of oscillating GNRs’ and Fig. 7b,c, accompanied by additional details in method section ‘Genotype maps of oscillatory synthetic GRNs’ and Supplementary Table 3).

For the specific questions:

(i) As we mention now in the main text, by parametrizing the dilution rate, the arabinose concentration and making an additional assumption on the unbinding of the dCas/sgRNA complex to the DNA (similar to the assumptions from Clamons & Murray, bioRxiv 225318; doi: <https://doi.org/10.1101/225318>), we could additionally account for dynamic phenotypes. In Fig 7b and Supplementary Table 3, we now show that most topologies containing a repressilator oscillate and this includes, in some cases, topologies that can also produce a stripe phenotype under different experimental setup at steady state. However, this analysis, as suggested by the reviewer, does not look at individual GRNs. Therefore, we considered both properties of 1) GREEN/ BLUE stripe (or not) at steady state and 2) oscillations (or not). By combining these two properties, we obtain 6 different phenotypes (No-Stripe/Oscillations, BLUE-Stripe/Oscillations, GREEN-Stripe/Oscillations, No-Stripe/No-Oscillations, BLUE-Stripe/No-Oscillations, GREEN-Stripe/No-Oscillations). Analyzing the transition frequencies

between neighboring genotypes for these phenotypes (similar to Fig 3d and Fig 7c), we observe that starting from a genotype that oscillates and can generate a stripe (e.g., BLUE-stripe/Oscillations phenotype) has an equal probability of maintaining this phenotype or losing the oscillatory property or both. In addition, there is a low probability of losing the oscillatory property but obtaining a new colored stripe or losing both properties completely. Concomitantly, if starting with a BLUE-stripe/No-Oscillations phenotype, at least with these assumptions on the model parametrizations, it is difficult to gain any kind of oscillatory property (Fig 2).

Figure 2: Transition frequencies starting with a specific phenotype. Phenotype abbreviations are No-Stripe/Oscillations: NoS/Osc, BLUE-Stripe/Oscillations: BS/Osc, GREEN-Stripe/Oscillations: GS/Osc, No-Stripe/No-Oscillations: NoS/NoOsc, BLUE-Stripe/No-Oscillations:BS/NoOsc, GREEN-Stripe/No-Oscillations: GS/NoOsc). We represent the median and in brackets the interquartile range.

(ii) The repression blue-green to green-blue would be considered as two mutations (according to the Hamming distance used, and because biologically, one repression would need to be lost, the other added).

Minor suggestions:

I very much appreciated the discussion of the size of mutational changes. However, I was thinking about this the whole time I was reading which was distracting. Could some of this be added to the introduction (near line 86), or the results paragraphs?

The second paragraph of the Results section has been modified according to the reviewer's suggestion, and it now reads as follows:

“Here, we modified topologies by adding or removing repression interactions, corresponding to a gain or loss of a sgRNA and/or its corresponding binding site. As for parameters, we modulated them in two ways: first, through the choice of three promoters (low, medium, high) that govern transcription of the nodes; and second, by employing four sgRNAs with different strengths. We also used two truncated versions (‘t4’, truncation of the four 5’ nucleotides) of the sgRNAs, which provides another way to tune repression strength⁵¹. Overall, changes involve differences ranging from 2-4nt (in the case of promoters and truncated sgRNAs) to 20nt (in the case of the sgRNAs and their binding sites). We consider each of these

modifications as a single mutational event (also see **Discussion**) and quantify relations between quantitative and qualitative changes using mathematical models.”

On my version, the font in Fig 1A is too small to read the promoter subscripts. This is a problem because it is needed to fully evaluate the logic of the network.

We believe that this was in part due to the poor quality of the images in the PDF generated by the submission tool. However, now we have increased the font size.

Fig. 2, I recommend labelling the GREEN-stripe and BLUE-stripe networks on the figure, not just in the legend. Some text to the right or left of the group should work.

Labels have been added to Fig. 2 as suggested by the reviewer.

Line 282, should this read “. . .either BLUE- or GREEN-stripe phenotypes”? instead of “both” “and”

We meant that the same topology provides access to two different phenotypes (BLUE- and GREEN-stripes) but not simultaneously: either one or the other, depending on the parameters. The sentence was indeed misleading and has been corrected: “In more detail, as suggested by our experimental data, mutual inhibition of green and blue nodes is required to enable a topology that can produce two different phenotypes – either BLUE or GREEN-stripes, depending on the parameters.”

Line 353, clarify what “substantially better” means

We have added the following clarification: “... perform substantially better (have higher robustness in both dimensions) ...”.

Line 418, a description of the logic of the oscillator would be nice. Why does it oscillate?

A description of the logic of the oscillator has been added: “The CRISPRator is composed of three nodes forming a negative feedback loop, each expressing a sgRNA that represses the next node in the loop. An increased level of any given node triggers a cascade of repression interactions that eventually bring its own levels down again leading to periodic oscillations.”

Line 520, “hamper” is a poor word choice. How does the analysis of epistasis hamper our understanding? I would think the analysis promotes our understanding.

The construction of the sentence was indeed somehow misleading. We have modified the sentence, which now says “More generally, epistasis profoundly complicates our understanding of how GRNs respond to mutations and is therefore studied extensively in different biological systems across scales⁴⁰.”

Reviewers' Comments:

Reviewer #1:

Remarks to the Author:

The authors have addressed all my queries in a satisfactory manner. I support publication.

Reviewer #2:

Remarks to the Author:

Thank you for your thorough response to my comments and questions. I have no further concerns. This is a very interesting contribution. Congratulations.